# Depletion of SMN protein in mesenchymal progenitors impairs the development of bone and neuromuscular junction in spinal muscular atrophy

**Sang-Hyeon Hann[1†], Seon-Yong Kim[2†], Ye Lynne Kim[1], Young-Woo Jo[1], Jong-Seol Kang[1], Hyerim Park[1], Se-Young Choi[2]\*, Young-Yun Kong[1]\***

[1]School of Biological Sciences, Seoul National University, Seoul, Republic of Korea; [2]Department of Physiology, Dental Research Institute, Seoul National University School of Dentistry, Seoul, Republic of Korea

**\*For correspondence:**
sychoi@snu.ac.kr (SYC);
ykong@snu.ac.kr (YYK)

[†]These authors contributed equally to this work

**Competing interest:** The authors declare that no competing interests exist.

**Abstract** Spinal muscular atrophy (SMA) is a neuromuscular disorder characterized by the deficiency of the survival motor neuron (SMN) protein, which leads to motor neuron dysfunction and muscle atrophy. In addition to the requirement for SMN in motor neurons, recent studies suggest that SMN deficiency in peripheral tissues plays a key role in the pathogenesis of SMA. Using limb mesenchymal progenitor cell (MPC)-specific SMN-depleted mouse models, we reveal that SMN reduction in limb MPCs causes defects in the development of bone and neuromuscular junction (NMJ). Specifically, these mice exhibited impaired growth plate homeostasis and reduced insulin-like growth factor (IGF) signaling from chondrocytes, rather than from the liver. Furthermore, the reduction of SMN in fibro-adipogenic progenitors (FAPs) resulted in abnormal NMJ maturation, altered release of neurotransmitters, and NMJ morphological defects. Transplantation of healthy FAPs rescued the morphological deterioration. Our findings highlight the significance of mesenchymal SMN in neuromusculoskeletal pathogenesis of SMA and provide insights into potential therapeutic strategies targeting mesenchymal cells for the treatment of SMA.

## eLife assessment

This **important** work by Hann et al. advances our understanding of the role of the survival motor neuron (SMN) protein in coordinating pathogenesis of the spinal muscular atrophy (SMA). The authors addressed many concerns raised by the reviewers, providing **convincing** evidence in terms of skeletal analyses not being able to satisfactorily elucidate SMN regulation of bone development.

## Introduction

The survival motor neuron (SMN) protein is a crucial component of the spliceosome complex and is essential for the proper function of all cell types (*Mercuri et al., 2022*). Deficiency in SMN protein disrupts the formation of spliceosome complexes, ultimately causing splicing defects in multiple genes. Mutations in the *SMN1* gene, which encodes the SMN protein, give rise to the neuromuscular disorder spinal muscular atrophy (SMA). SMA is characterized by neuromuscular junctions (NMJs) disruption, muscular atrophy, and alpha motor neuron loss (*Mercuri et al., 2022*; *Burghes and Beattie, 2009*). The severity of the disease in humans correlates with the copy number of *SMN2*, which is a paralog of *SMN1* in humans. *SMN2* primarily produces less functional exon 7-deleted SMN protein and rarely generates a limited quantity of functional full-length SMN protein via alternative

splicing. SMA patients are classified into types 0 through 4 based on the severity and timing of disease onset, predominantly determined by the number of copies of the *SMN2* gene they possess. More than 50% of SMA patients are categorized as type 1, characterized by muscle defects with proximal muscle atrophy during infancy, eventually resulting in death within a few years (*Mercuri et al., 2022*).

Previous studies suggest that the onset of SMA is mainly attributed to SMN loss in motor neurons (*Monani et al., 2000*; *Burghes and Beattie, 2009*). Nevertheless, motor neuron-specific SMN deficiency in the SMA mouse model exhibits relatively mild phenotypes compared to whole-body SMA mouse models (*Park et al., 2010*; *McGovern et al., 2015*). Furthermore, restoring SMN to motor neurons in SMA mouse models result in only partial rescue in lifespan and neuromuscular defects (*Passini et al., 2010*; *Martinez et al., 2012*; *McGovern et al., 2015*; *Besse et al., 2020*). Systemic administration of antisense oligonucleotide (ASO), which corrects *SMN2* splicing to restore SMN expression, significantly prolongs survival compared to central nervous system (CNS) administration (*Hua et al., 2011*). In the mouse treated with a systemically delivered ASO, blocking the effect of ASO in the CNS by a complementary decoy did not have any detrimental effect on survival, motor function, or NMJ integrity (*Hua et al., 2015*). These studies suggest that peripheral SMN plays a crucial part in SMA pathology. Overall, investigating the impacts of SMN depletion in peripheral tissues is critical for alleviating neuromuscular impairments and increasing life expectancy in SMA.

In SMA patients, bone growth retardation has been observed (*De Amicis et al., 2021*; *Kipoğlu et al., 2019*; *Hensel et al., 2020*). Studies using whole-body SMA mouse models have revealed that this is caused by diminished growth plate chondrocyte density and endochondral ossification defects, independent of muscle atrophy (*Hensel et al., 2020*). However, it is still unclear whether these defects result from the SMN ablation in bone-forming cells, or from a decline in liver-derived insulin-like growth factor (IGF) in SMA patients and mice (*Hua et al., 2011*; *Yesbek Kaymaz et al., 2016*). In severe SMA mice, the serum levels of IGF decreased by approximately 60% or became undetected (*Hua et al., 2011*; *Murdocca et al., 2012*). The decreased serum IGF levels were attributed to decreased expression of liver genes, including *Igf1*, IGF binding, and ternary complex protein gene *Igfals* and *Igfbp3*. The previous studies suggest that the liver is the primary origin of systemic IGF, as demonstrated by the liver-specific deletion of *Igf1* and the knockout of *Igfals*, which is mostly expressed in the liver (*Yakar et al., 1999*; *Yakar et al., 2002*). The double KO mice exhibited a 90% decrease in serum IGF levels and displayed a phenotype of shortened femur length and growth plates. It is thus possible that the decrease in serum IGF levels, resulting from reduced liver IGF pathway genes in SMA mice, has also played a role in the observed bone growth defect.

Mesenchymal progenitor cells (MPCs) derived from the lateral plate mesoderm (LPM) differentiate into various types of limb mesenchymal cells, including bone, cartilage, and intramuscular mesenchymal cells like fibro-adipogenic progenitors (FAPs) (*Nassari et al., 2017*). Recent studies have revealed the role of FAPs in skeletal muscle homeostasis, as reducing the number of FAPs resulted in diminished muscle regeneration capacity, long-term muscle atrophy, and NMJ denervation (*Wosczyna et al., 2019*; *Uezumi et al., 2021*). Our latest research demonstrated that FAP-specific deficiency of *Bap1*, one of the deubiquitinases, leads to NMJ defects (*Kim et al., 2022*). These recent findings raise the possibility that FAPs may have a specific role in the pathogenesis of neuromuscular diseases such as SMA. However, it has not been studied if depletion of SMN in FAPs can lead to SMA-like neuromuscular pathology.

In this study, we crossed a limb MPC-specific Cre mouse with a floxed *Smn1* exon 7 mouse carrying multiple copies of the human *SMN2* gene, allowing us to examine the impact of mesenchymal SMN reduction on SMA pathogenesis. As a result, the mutant mice showed skeletal growth abnormalities and local IGF signaling defects in the growth plate. In addition, our findings indicate that the SMN reduction in FAPs, similar to the extent of severe SMA, causes altered NMJ development.

## Results

### Bone growth restriction and growth plate defects caused by MPC-specific SMN depletion

To investigate the effects of SMN reduction within MPCs in SMA pathogenesis, we crossed *Smn1^{f7/f7}* mice, which possess loxP sites flanking exon 7 of the *Smn1* gene (*Frugier et al., 2000*), with *Prrx1^{Cre}* mice. This produced *Smn1^{ΔMPC}* mice (*Prrx1^{Cre}*; *Smn1^{f7/f7}*) that lacked the *Smn1* gene specifically in

*Prrx1^{Cre}*-expressed limb MPCs that give rise to bone, cartilage, and FAPs (*Logan et al., 2002*; *Leinroth et al., 2022*). To ascertain whether mutant mice carrying the *SMN2* gene, like SMA patients, present pathological phenotypes, we additionally generated *SMN2* 2-copy *Smn1^{ΔMPC}* (*Prrx1^{Cre}*; *Smn1^{f7/f7}*; *SMN2^{+/+}*) and *SMN2* 1-copy *Smn1^{ΔMPC}* (*Prrx1^{Cre}*; *Smn1^{f7/f7}*; *SMN2^{+/0}*) mice. Control littermates that lacked *Prrx1^{Cre}* were used as controls for comparison.

*SMN2* 0-copy *Smn1^{ΔMPC}* mice died within 24 hr after birth. Regions where limbs should have formed at E18.5 only had rudimentary limb structures (*Figure 1—figure supplement 1A*). Furthermore, since the upper head bone did not cover the brain, it was directly attached to the skin and protruded. These observations can be attributed to *Prrx1^{Cre}*-mediated SMN deletion in the LPM-derived limb MPCs and the craniofacial mesenchyme, which is accountable for the formation of calvarial bone (*Wilk et al., 2017*). To investigate whether the lack of SMN proteins in MPCs is responsible for bone development abnormalities, we performed alcian blue and alizarin red staining on E18.5 *SMN2* 0-copy *Smn1^{ΔMPC}* mutants to analyze the structure of bones and cartilage. The appendages displayed restricted bone and cartilage formations, with scarcely discernible femur and tibia (*Figure 1—figure supplement 1B, C*). In the cranial region, there was an absence of both cartilage and bone at the location of the calvarial bone, with the parietal bone entirely missing and partially absent frontal bone (*Figure 1—figure supplement 1D*). Additionally, the sternum, which is one of the bones originating from the LPM (*Sheng, 2015*), was shorter than the control (*Figure 1—figure supplement 1E*).

The *SMN2* 2-copy *Smn1^{ΔMPC}* mice, carrying two homologous *SMN2* genes, did not show any discernible differences from the *Prrx1^{Cre}*-negative control littermates into adulthood. However, the *SMN2* 1-copy *Smn1^{ΔMPC}* mice exhibited reduced body size and shorter limb length compared to the *SMN2* 1-copy control (*Smn1^{f7/f7}*; *SMN2^{+/0}*). To assess postnatal bone growth defects observed in SMA patients and mouse models, we conducted micro-computed tomography (micro-CT) analysis on femurs obtained from postnatal day 14 (P14) *SMN2* 1-copy *Smn1^{ΔMPC}* and *SMN2* 2-copy *Smn1^{ΔMPC}* mice (*Figure 1A*). The 3D reconstruction image showed that the *SMN2* 1-copy mutant femur was smaller than the WT control and *SMN2* 2-copy mutant, and secondary ossification center is denied. The longitudinal virtual section view displayed reduced trabecular bone in the *SMN2* 1-copy mutant femur. CT analysis data showed that *SMN2* 1-copy mutants exhibited reduced femur diaphysis length, diameter, and trabecular bone volume compared to the control group, indicating growth plate-dependent endochondral ossification defects (*Figure 1B–D*). We then examined femoral bone thickness and diaphyseal bone mineral density (BMD) to determine whether mineralization was normal after bone formation. The thickness of the bone in the mutants did not differ significantly from the control, suggesting that bone mineralization was intact (*Figure 1E* and *Figure 1—figure supplement 2A*). Unexpectedly, BMD slightly increased in *SMN2* 1-copy mutants compared to the control group (*Figure 1—figure supplement 2B*). To assess the impact of osteoclasts and osteoblasts on diaphysis cortical bone mineralization, we utilized *Itgb3* immunofluorescence as the osteoclast marker and toluidine blue staining for imaging bone-attached osteoblasts (*Romeo et al., 2019*; *Colaianni et al., 2015*). Osteoclast and osteoblast density did not significantly differ between the *SMN2* 1-copy mutant and the control (*Figure 1—figure supplement 2C–E*). The higher BMD may be attributed to greater mechanical stress caused by the shorter femur supporting the weight of the body, consistent with prior research indicating that elevated mechanical force leads to higher BMD in the femur (*Hoxha et al., 2014*; *Ike et al., 2015*). Nevertheless, the decrease in bone growth without apparent deterioration in bone mineralization of the femur of *SMN2* 1-copy *Smn1^{ΔMPC}* mutants is consistent with findings from the whole-body SMA mouse model (*Hensel et al., 2020*). Collectively, these results suggest that mice carrying low copies of *SMN2*, with the *Smn1* gene specifically deleted in MPCs, exhibit bone growth abnormalities, while osteoblast and osteoclast populations show no obvious defects based on our preliminary analyses.

Various bone-forming cells originating from the LPM were known to contribute to bone formation, such as growth plate chondrocytes and osteoblasts. In *SMN2* 1-copy *Smn1^{ΔMPC}* mutants, SMN may be deleted in these cells, suggesting that they play a role in the bone growth abnormalities observed in 2-week-old mice. Previous researchers revealed that primary osteoblasts from a severe SMA mouse model did not display notable differences from controls in an in vitro ossification test. And they did not observe any differences in bone voxel density and bone thickness in femurs at P3 severe SMA mice. This is supported by the absence of any bone thickness or BMD defects in the *SMN2* 1-copy mutant (*Figure 1E* and *Figure 1—figure supplement 2A, B*), and the unimpaired osteoblast population

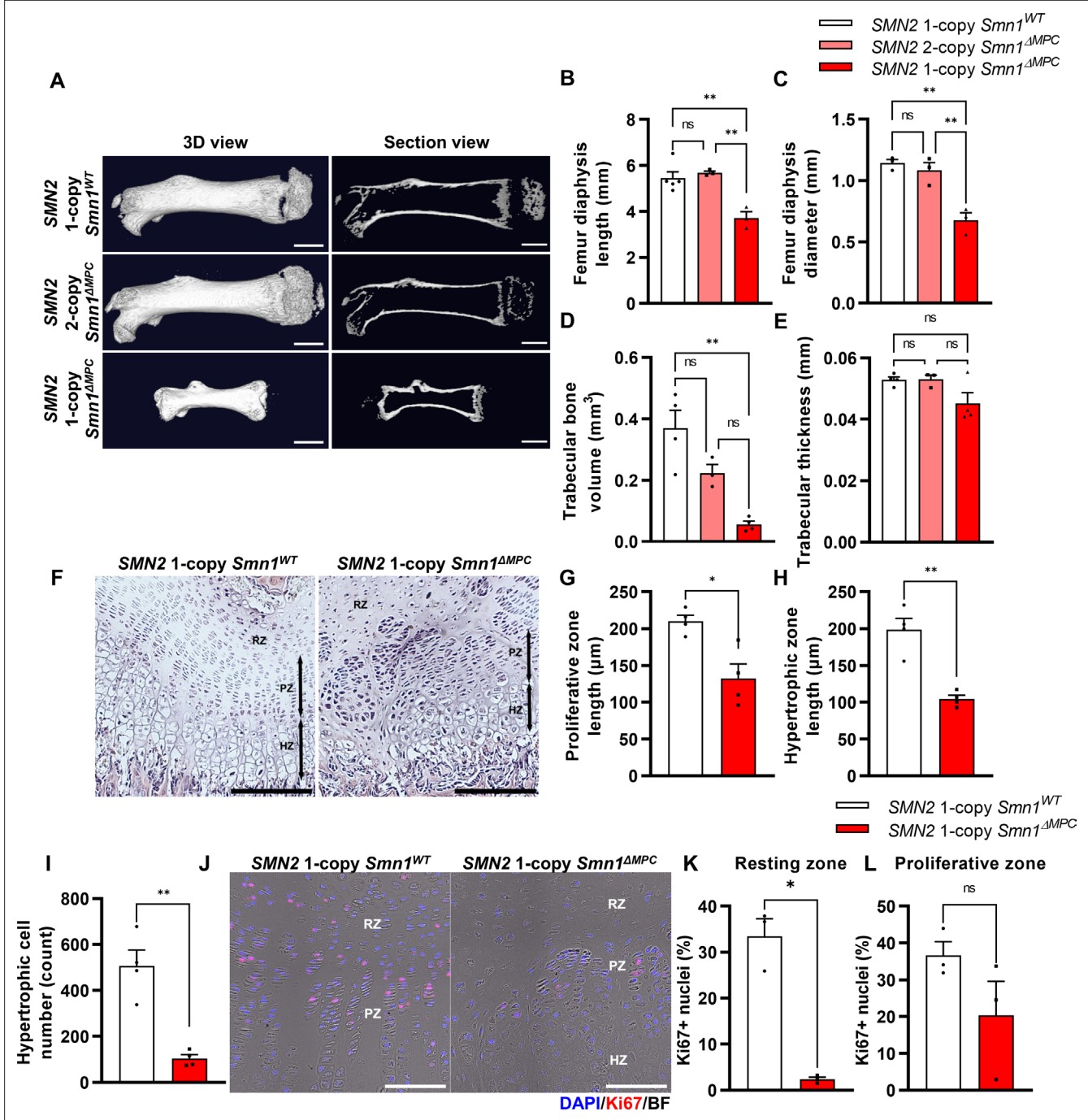

**Figure 1.** Skeletal growth abnormalities and altered growth plate homeostasis in *SMN2* 1-copy *Smn1^ΔMPC^* mice. (**A**) Representative 3D images and longitudinal section view of the ossified femur bone. Scale bars, 1 mm. (**B, C**) *SMN2* 1-copy mutant's femurs showed reduced growth in diaphysis length and diameter, and (**D**) decreased trabecular bone volume. (**E**) Trabecular bone thicknesses were not significantly different between the control and mutant groups. The micro-computed tomography (micro-CT) analysis was performed in femur diaphysis and metaphysis from *SMN2* 1-copy *Smn1^ΔWT^*, *SMN2* 2-copy, and 1-copy *Smn1^ΔMPC^* mice at P14. One-way analysis of variance (ANOVA) with Tukey's post hoc test, n = 3–5 mice in each genotype (**B–E**). (**F**) Representative images of hematoxylin and eosin (H&E) staining in the distal femur growth plate of control and mutant mice with 1 copy of *SMN2* at P14. Scale bars, 100 μm. Resting zone (RZ), hypertrophic zone (HZ), and proliferative zone (PZ). (**G–I**) Indicated by black arrows, the HZ and PZ lengths were reduced in *SMN2* 1-copy *Smn1^ΔMPC^* mice, and the hypertrophic cell number in a section of the 1-copy mutant was decreased (n = 4 mice in each genotype; unpaired *t*-test with Welch's correction). (**J**) Representative images of Ki67 immunostaining in the distal femur growth plate of control and mutant mice with 1 copy of *SMN2* at P14. Scale bars, 100 μm, and (**K**) decreased Ki67+ percentage in resting zone chondrocytes. (**L**) Ki67+ percentage in the proliferative zone was not significantly different between the control and mutant groups. n = 3 mice in each genotype; unpaired *t*-test with Welch's correction (**K–L**). ns: not significantly different. *p < 0.05; **p < 0.01. Error bars show standard error of the mean (SEM).

*Figure 1 continued on next page*

*Figure 1 continued*

The online version of this article includes the following figure supplement(s) for figure 1:

**Figure supplement 1.** Growth defects in the *Prrx1*-lineage bone of *SMN2* 0-copy *Smn1^ΔMPC^* mice.

**Figure supplement 2.** Osteoclasts and osteoblasts were undisturbed in *SMN2* 1-copy *Smn1^ΔMPC^* mice.

---

(*Figure 1—figure supplement 2E*). Thus, we conclude that the bone growth abnormalities observed in the 2-week-old *SMN2* 1-copy *Smn1^ΔMPC^* mutant are due to impaired endochondral ossification.

To determine whether bone growth defects in *SMN2* 1-copy *Smn1^ΔMPC^* mutants arise from disrupted chondrocyte homeostasis at growth plates, we stained the femur distal growth plate of P14 mice with hematoxylin and eosin (H&E; *Figure 1F*). In line with earlier findings in whole-body SMA mice (*Hensel et al., 2020*), *SMN2* 1-copy *Smn1^ΔMPC^* mice exhibited shorter proliferative and hypertrophic zones compared to control mice (*Figure 1G, H*). Additionally, there was a significant reduction in the number of chondrocytes in the hypertrophic zone (*Figure 1I*). To investigate the decrease in chondrocyte proliferation and subsequent reduction in the proliferative and hypertrophic zone, we stained the proliferation marker Ki67 in the growth plate of both *SMN2* 1-copy control and mutant samples (*Figure 1J*). We then quantified the percentage of Ki67+ nuclei in the resting and proliferative zones (*Figure 1K, L*). Although there was no significant difference observed in the Ki67+ percentage of proliferative zone chondrocytes in the proceeding proliferation state, there was an absolute reduction in resting zone chondrocyte proliferation. The decreased proliferation rate in the resting zone could have impeded the transition to the proliferative zone. Our data indicate that adequate expression of SMN is essential for the homeostasis of chondrocytes at growth plates.

## Disruption of chondrocyte-derived IGF signaling in *SMN2* 1-copy *Smn1^ΔMPC^* mutants

The proliferation and differentiation of growth plate chondrocytes are regulated by systemic IGF (*Shim, 2015*; *Karimian et al., 2011*; *Racine and Serrat, 2020*). Previous research suggested that a key factor contributing to the pathological phenotype in SMA is the lowered expression of the *Igf1/Igfbp3/Igfals* genes, which produce IGF and IGF-carrying proteins, in the liver (*Hua et al., 2011*; *Murdocca et al., 2012*). As the IGF pathway proteins are downregulated in whole-body SMA mice, the bone growth defects observed in the mice have sparked debate as it remains unclear whether they are due to cell-autonomous defects by bone-forming cells' SMN reduction, or the low liver-derived IGF level (*Hua et al., 2011*; *Tsai et al., 2014*; *Deguise et al., 2021*; *Hensel et al., 2020*).

To clarify this issue, we used *Smn1^ΔMPC^* mutant mice, which enabled us to investigate the effect of SMN depletion in bone-forming cells such as chondrocytes on bone growth, while circumventing the impact of the endocrine signal by *Prrx1*-negative organ. To investigate the impact of IGF signaling on growth plate chondrocytes, we employed immunofluorescence to evaluate the percentage of p-AKT-positive cells activated by the IGF–PI3K–AKT pathway in both *SMN2* 1-copy control and mutant femur distal growth plate (*Figure 2A*). Intriguingly, the percentage of p-AKT+ cells was significantly decreased in resting zone chondrocytes, but not in the proliferative zone, which aligns with the Ki67+ percentage (*Figure 2B, C*). Expectedly, the liver's mRNA expression of IGF pathway genes, reported to be decreased in SMA mouse models and patients (*Hua et al., 2011*; *Murdocca et al., 2012*; *Deguise et al., 2021*; *Sahashi et al., 2013*), showed no difference when comparing controls to *SMN2* 1-copy *Smn1^ΔMPC^* mutants (*Figure 2D*). These findings indicate that the growth plate's proliferation and hypertrophy in *SMN2* 1-copy mutants are affected by impairments in another AKT upstream signal rather than by liver-secreted systemic IGF.

There are reports indicating that local IGF expression plays a crucial role in bone development through the growth plate, in addition to circulating IGF (*Hallett et al., 2019*; *Racine and Serrat, 2020*). While most *Igf1*-null mice died before birth and had smaller tibial lengths than normal, liver-specific *Igf1*-deleted mice did not experience significant changes in body length or tibial length during postnatal growth, despite a 75% reduction in serum IGF-1 levels (*Baker et al., 1993*; *Yakar et al., 1999*). This indicates that local IGF in the growth plate is crucial for endochondral ossification, in addition to serum IGF. The chondrocyte-specific Igf1 knockout mouse demonstrated a reduction in postnatal body and femur length, and the chondrocyte-specific *Igf1r* knockout mouse demonstrated a significant reduction in bone growth, as well as a decrease in both growth plate proliferative and

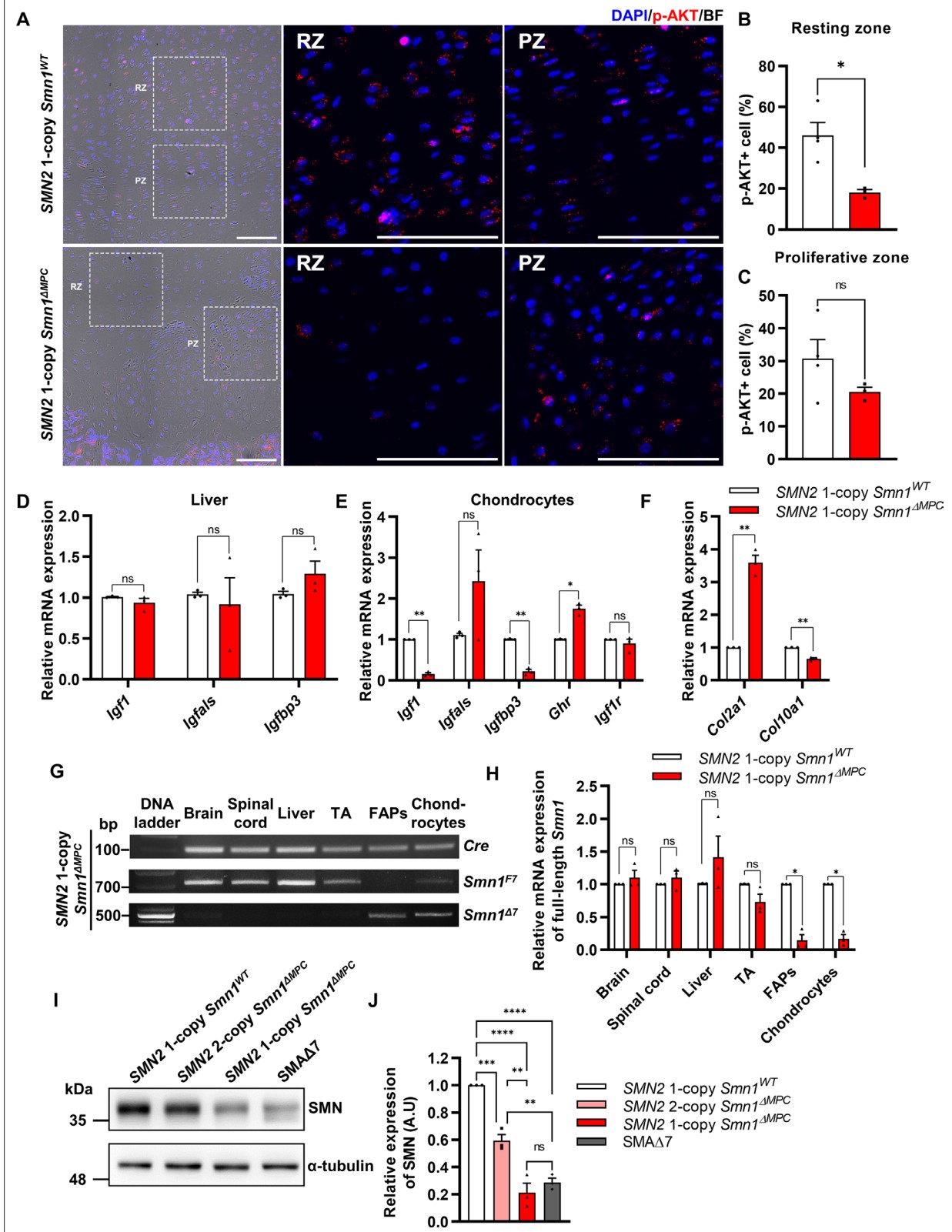

**Figure 2.** Decreased chondrocyte-derived IGF–AKT axis by limb mesenchymal cell-specific survival motor neuron (SMN) depletion in *SMN2* 1-copy *Smn1^ΔMPC^* mice. (**A**) Representative images of p-AKT immunostaining in distal femur growth plate from mice at P14. Scale bars, 100 μm. (**B, C**) The p-AKT-positive percentage was decreased in the resting zone chondrocytes, not in the proliferative zone (*n* = 3–4 mice in each genotype; unpaired *t*-test with Welch's correction). (**D**) Relative IGF axis mRNA expression in the livers of *SMN2* 1-copy control and mutant mice. The IGF pathway genes

*Figure 2 continued on next page*

*Figure 2 continued*

showed no difference when comparing controls to *SMN2* 1-copy *Smn1^ΔMPC^* mutants. (**E, F**) Relative IGF axis and chondrocyte differentiation marker mRNA expression in the chondrocytes of *SMN2* 1-copy control and mutant mice. The *Igf1, Igfbp3,* and hypertrophic marker *Col10a1* expression were decreased in *SMN2* 1-copy *Smn1^ΔMPC^* mutants. *n* = 3 mice in each genotype; unpaired *t*-test with Welch's correction (**D–F**). (**G**) Representative images of genomic PCR analysis from *SMN2* 1-copy *Smn1^ΔMPC^* mice tissues at P21. (**H**) Quantitative reverse transcription polymerase chain reaction (qRT-PCR) analysis from tissues of *SMN2* 1-copy *Smn1^WT^* and *Smn1^ΔMPC^* mice at P21 (*n* = 3 mice in each genotype; unpaired *t*-test with Welch's correction). Deletion of *Smn1* exon 7 was detected only in limb mesenchymal cells using genomic PCR (**G**) and full-length *Smn1* mRNA expression (**H**). (**I**) Representative images of western blot analysis in cultured fibro-adipogenic progenitors (FAPs). SMN protein in FAPs of *SMN2* 1-copy *Smn1^ΔMPC^* mice exhibited a decrease comparable to that observed in the SMAΔ7 mice. (**J**) Relative SMN levels in cultured FAPs of the controls and mutants (*n* = 3 mice in each genotype; one-way analysis of variance (ANOVA) with Tukey's post hoc test). ns: not significantly different. *p < 0.05; **p < 0.01; ***p < 0.001; ****p < 0.001. Error bars show standard error of the mean (SEM).

The online version of this article includes the following source data for figure 2:

**Source data 1.** Original file for the gel electrophoresis of genomic PCR in *Figure 2G* (*Cre, Smn1^F7^, Smn1^Δ7^*) and western blot analysis in *Figure 2I* (anti-alpha-tubulin, anti-SMN).

**Source data 2.** PDF containing *Figure 2G, I* and original scans of the PCR and western blot with highlighted bands and sample labels.

hypertrophic zone (*Govoni et al., 2007*; *Wang et al., 2011*). A recent study revealed that resting zone chondrocytes in the growth plate serve as a major source of local IGF and activate the p-AKT pathway via autocrine and paracrine IGF signaling (*Oichi et al., 2023*). The study further revealed that cells that constitute bone and bone marrow, apart from chondrocytes, do not express *Igf1*, making chondrocytes the solitary source of local IGF. These suggest that the growth plate defects and the reduction of resting zone AKT phosphorylation in *SMN2* 1-copy mutants may be due to chondrocyte-secreted IGF deficiency. To confirm this hypothesis, we evaluated the expression of IGF-related genes in chondrocytes from both *SMN2* 1-copy control and mutant femur (*Figure 2E*). Indeed, *Igf1* and *Igfbp3* were greatly depleted in *SMN2* 1-copy mutant chondrocytes. It is hypothesized that the increased presence of *Ghr* may be in response to the reduction of *Igf1*. As IGF directly causes chondrocyte hypertrophy (*Wang et al., 1999*), we assess the mRNA expression of chondrocyte hypertrophic marker *Col10a1* and undifferentiated chondrocyte marker *Col2a1* (*Figure 2F*). The results show that *Col10a1* is decreased, while *Col2a1* is increased in the mutant. In *Figure 1H, I*, the hypertrophic cell reduction may be caused by a low local IGF level. Therefore, depletion of local IGF in the growth plate of *SMN2* 1-copy mutants may hinder chondrocyte progression to proliferation and hypertrophy, leading to aberrations in endochondral ossification. In *Figures 1L and 2C*, it is possible that the serum IGF could affect the remnant proliferative and p-AKT-positive cells in the growth plate via the vascularization of bone marrow. Based on these findings, it was concluded that deprivation of SMN in chondrocytes leads to a decrease in local IGF signaling, which affects growth plate homeostasis.

## Mesenchymal cell-specific SMN reduction similar to severe SMA mouse model in *SMN2* 1-copy *Smn1^ΔMPC^* mutants

To confirm the specific deletion of *Smn1* in limb mesenchymal cells, including chondrocytes and FAPs, we performed quantitative reverse transcription polymerase chain reaction (qRT-PCR) for full-length *Smn1* mRNA expressed by undeleted *Smn1* allele. Our findings indicate that full-length *Smn1* mRNA expression in the brain, liver, skeletal muscle, and spinal cord of *SMN2* 1-copy *Smn1^ΔMPC^* mice at post-natal day 21 (P21) was similar to that of control mice, while it was significantly reduced in isolated FAPs and chondrocytes (*Figure 2H*). Additionally, we confirmed the presence of the *Smn1^Δ7^* variant, which is the exon 7-deleted form of *Smn1* created by cre-lox-mediated recombination, in both FAPs and chondrocytes by conducting genomic PCR on the *SMN2* 1-copy *Smn1^ΔMPC^* mutant (*Figure 2G*). Since SMN was not downregulated in the tissues other than limb mesenchymal cells, non-mesenchymal cells were ruled out from being responsible for the phenotype observed in *Smn1^ΔMPC^* mutants.

Among our *Smn1^ΔMPC^* models, mice with two copies of *SMN2* exhibit similar bone development parameters as control mice without SMN deletion. However, mice with one copy of *SMN2* display bone pathological defects akin to SMA mouse models. This may be because the quantity of human full-length SMN protein produced by *SMN2* 2-copy, was sufficient to sustain SMN complex function in the *SMN2* 2-copy *Smn1^ΔMPC^* mutants, despite the absence of functional mouse SMN protein in limb mesenchymal cells. On the contrary, due to the lower expression of full-length SMN compared to *SMN2* 2-copy mutants, the SMN complex may not function properly in *SMN2* 1-copy mutant cells. We

confirmed this by comparing the amount of full-length SMN protein in isolated FAPs from the hind-limbs of control, $Smn1^{\Delta MPC}$ mutants, and a severe SMA mouse model ($Smn1^{-/-}$; $SMN2^{+/+}$; $SMN\Delta7^{+/+}$; SMAΔ7 mutants) (*Figure 2I*). The data show that *SMN2* 1-copy $Smn1^{\Delta MPC}$ mutants exhibited ~80% reduction in SMN levels compared to the control group. The level of SMN protein in *SMN2* 1-copy $Smn1^{\Delta MPC}$ mutants was similar to that in SMAΔ7 mutants. Conversely, *SMN2* 2-copy mutants display a decrease of approximately 40% in SMN protein levels compared to the control (*Figure 2J*). The moderately reduced expression of SMN is adequate to support regular bone development in *SMN2* 2-copy mutants. Taken together, our findings indicate that the reduction of mesenchymal SMN to levels comparable to that of the severe SMA mouse model causes SMA-like bone pathology in the *SMN2* 1-copy mutant.

## Abnormal NMJ maturation in *SMN2* 1-copy *Smn1*$^{\Delta MPC}$ mutants

To investigate whether disabling SMN in FAPs results in SMA-like neuromuscular impairments, we assessed if NMJ phenotypes observed in SMA mouse models also occur in *SMN2* 1-copy $Smn1^{\Delta MPC}$ mutants. Both severe and mild SMA mouse models exhibit impaired NMJ maturation markers, including plaque-like morphology of acetylcholine receptor (AChR) clusters, neurofilament (NF) vari-cosities, and poor terminal arborization (*Kong et al., 2009*; *Martinez et al., 2012*; *Monani et al., 2003*; *Kariya et al., 2008*). To evaluate the impact of SMN deficiency in FAPs on NMJ maturation, we evaluated the NMJ maturation markers in the tibialis anterior (TA) muscles of control and *SMN2* 1-copy $Smn1^{\Delta MPC}$ mutant mice at P21, a time when NMJ maturation is in progress (*Figure 3A*). Our examination revealed the presence of NF varicosities in *SMN2* 1-copy mutants as compared with control mice (*Figure 3A, D*). Additionally, the number of nerve branches was decreased and half of the total NMJs were poorly arborized in *SMN2* 1-copy mutants (*Figure 3B, C*). These presynaptic alterations are specific phenotypes in neurogenic atrophy-like SMA. Unlike neurogenic atrophy, phys-iologic atrophy shows no differences in presynaptic morphology, such as nerve branching (*Deschenes et al., 2006*). This suggests that the NMJ phenotypes observed in *SMN2* 1-copy $Smn1^{\Delta MPC}$ mutant mice are not caused by decreased muscle size and activity resulting from bone growth abnormalities. The morphology of AChR clusters shows that the mutants have more immature plaque-like NMJs than the controls' pretzel-like structure (*Figure 3E*). Therefore, these findings indicate that *SMN2* 1-copy $Smn1^{\Delta MPC}$ mutants exhibit NMJ maturation abnormalities common in SMA mouse models.

## Undisturbed NMJ formation in neonatal *SMN2* 1-copy *Smn1*$^{\Delta MPC}$ mutants

To determine whether any NMJ defects were present prior to juvenile NMJ maturation in *SMN2* 1-copy $Smn1^{\Delta MPC}$, we examined NMJ formation in *SMN2* 1-copy $Smn1^{\Delta MPC}$ mice at the neonatal stage on postnatal day 3. We evaluated the AChR and nerve terminal areas to assess post- and presynaptic development, respectively (*Figure 3F*). Measurements of AChR cluster size indicated no differences between control and *SMN2* 1-copy $Smn1^{\Delta MPC}$ mice (*Figure 3G*). However, the area of AChR covered by nerve terminals was slightly larger in *SMN2* 1-copy $Smn1^{\Delta MPC}$ (*Figure 3H*). We have no reasonable explanation for why the coverage is higher in the mutant. However, there does not appear to be abnormal development of the NMJ in the mutant, at least until the neonatal period. Therefore, we reasoned that *SMN2* 1-copy $Smn1^{\Delta MPC}$ mutants began to exhibit deterioration in the NMJ maturation during the juvenile stage, following the intact neonatal development of NMJ.

## Aberrant NMJ morphology in the adult *SMN2* 1-copy *Smn1*$^{\Delta MPC}$ mice

To evaluate the organization of NMJ after the conclusion of postnatal NMJ development, consid-ering mesenchymal SMN expression, we examined NMJ morphology in the TA muscle of control, *SMN2* 1-copy, and 2-copy $Smn1^{\Delta MPC}$ mice at postnatal day 56 (P56). Our analysis revealed that presynapses were fragmented in *SMN2* 1-copy mutants, resulting in a bouton-like morphology, in contrast to the control and *SMN2* 2-copy $Smn1^{\Delta MPC}$ mice (*Figure 4A*). In *SMN2* 1-copy $Smn1^{\Delta MPC}$ mice, a twofold presynaptic fragmentation compared to the control was quantified, demonstrating nerve terminal shrinkage (*Figure 4B*). Additionally, NF ends displayed more severe varicosities than at P21 and were only connected to the proximal nerve by very thin NF, unlike control and *SMN2* 2-copy mice (*Figure 4C*). Remarkably, numerous presynaptic islands formed in *SMN2* 1-copy $Smn1^{\Delta MPC}$ mice through the merging of fragmented presynapses and NF varicosity. In *SMN2* 1-copy mutants, AChR

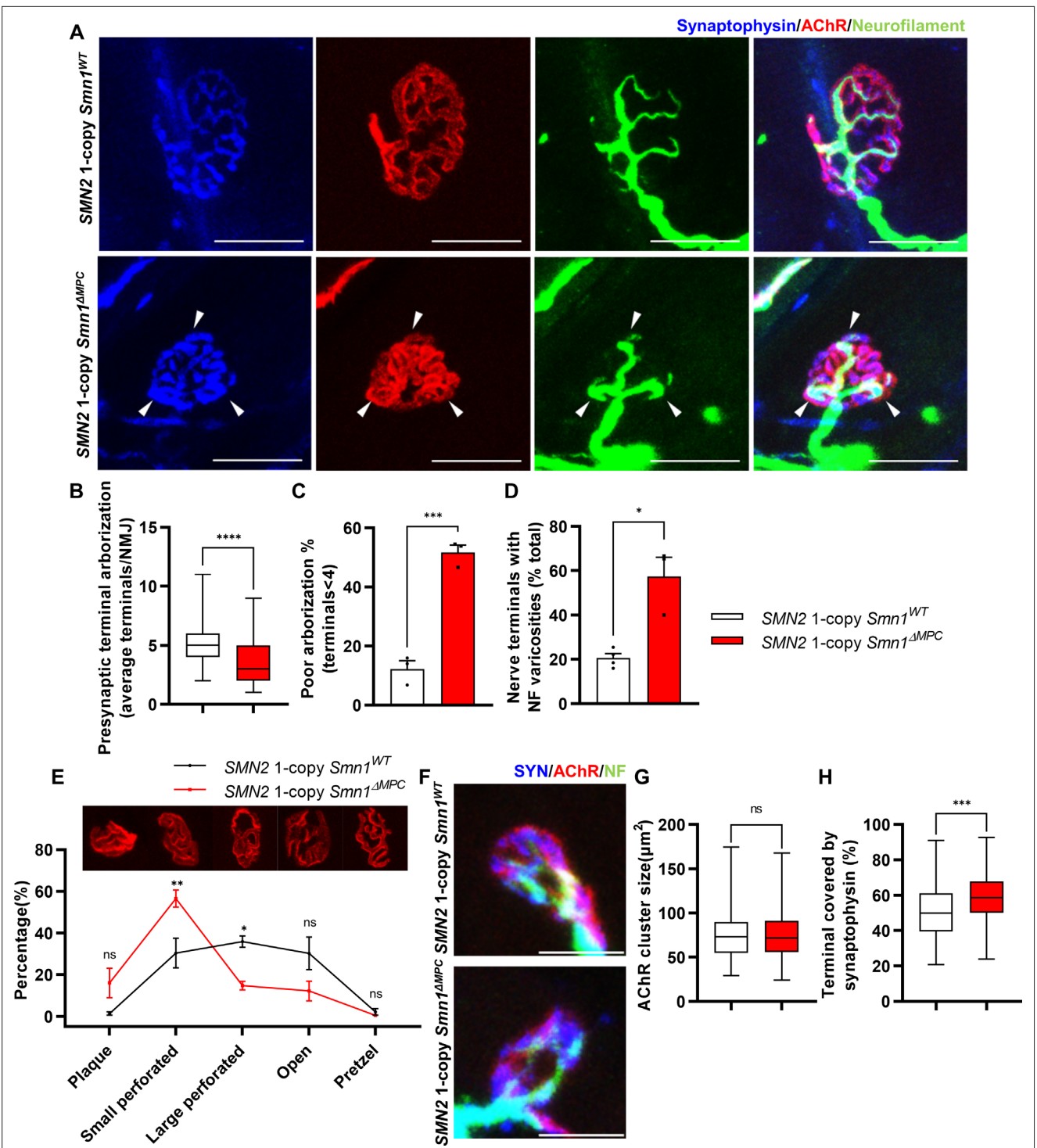

**Figure 3.** Aberrant postnatal neuromuscular junction (NMJ) maturation in *SMN2* 1-copy *Smn1^ΔMPC* mice. (**A**) Immunostaining of NMJs in TA muscle of *SMN2* 1-copy *Smn1^WT* and *Smn1^ΔMPC* mice at P21 with anti-NF (green), anti-synaptophysin (blue), and α-Btx staining acetylcholine receptor (AChR; red). Scale bars, 20 μm. The confocal images of NMJs showed decreased presynaptic terminal branching and the existence of nerve terminal varicosities that were enlarged with neurofilament (NF; indicated by arrowheads) in the mutant. (**B**) The NMJs of the *SMN2* 1-copy mutant exhibited a significant decrease in presynaptic terminal arborization and (**C**) an increased percentage of poorly arborized NMJs (*n* = 3 mice in each genotype; unpaired *t*-test with Welch's correction). (**D**) The percentage of NMJs exhibiting NF varicosities was higher in the *SMN2* 1-copy mutant group than in the control group (*n* = 3–4 mice in each genotype; unpaired *t*-test with Welch's correction). (**E**) For quantification of the NMJ maturation stage, we classified NMJs into five distinct developmental stages (Plaque: plaque-shaped endplate without any perforation; Small perforated: plaque-shaped endplate with small

*Figure 3 continued on next page*

*Figure 3 continued*

perforations; Large perforated: plaque-shaped endplate with large perforations; Open: C-shaped endplate; Pretzel: pretzel-like shaped endplate) and then compared the frequency patterns of *SMN2* 1-copy control and mutant mice (*n* = 3 mice in each genotype; two-way analysis of variance (ANOVA) with Tukey's post hoc test). The NMJs of *SMN2* 1-copy mutants displayed plaque-like shapes, indicating that they were in the immature stage. (**F**) Immunostaining of NMJs in TA muscle of *SMN2* 1-copy *Smn1*$^{WT}$ and *Smn1*$^{\Delta MPC}$ mice at P3 with anti-NF (green), anti-synaptophysin (blue), and α-Btx staining AChR (red). Scale bars, 10 μm. (**G**) There were no significant differences in AChR cluster size between the *SMN2* 1-copy control and mutant at P3 (*n* = 3–4 mice in each genotype; unpaired *t*-test with Welch's correction). (**H**) The ratio of the Synaptophysin area to the AChR area in NMJ was slightly higher in the *SMN2* 1-copy mutant at P3 (*n* = 3–4 mice in each genotype; unpaired *t*-test with Welch's correction). ns: not significantly different. *p < 0.05; **p < 0.01; ***p < 0.001; ****p < 0.0001. All box-and-whisker plots show the median, interquartile range, minimum, and maximum. For the box-and-whisker plots, range bars show minimum and maximum (**B, G, H**). For the bar and line graph, error bars show standard error of the mean (SEM) (**C–E**).

clusters displayed fragmented grape-shaped morphology that overlapped with nerve terminals, whereas control and *SMN2* 2-copy mice displayed pretzel-like structures (*Figure 4D*). These results suggest that defects in adult NMJ morphology occur when mesenchymal SMN protein is reduced to the extent of the *SMN2* 1-copy *Smn1*$^{\Delta MPC}$ mutants.

## Presynaptic neurotransmission alteration in *SMN2* 1-copy *Smn1*$^{\Delta MPC}$ mutants

To investigate whether the morphologically aberrant NMJs of *SMN2* 1-copy *Smn1*$^{\Delta MPC}$ mice have functional impairments, we isolated hindlimb extensor digitorum longus (EDL) muscles from P56 mice and conducted electrophysiological recording. We incubated the muscles with μ-conotoxin, which selectively inhibits muscle voltage-gated Na$^+$ channels, prevented the induction of muscle action potential (*Ling et al., 2010*; *Zanetti et al., 2018*), and recorded the Miniature endplate potential (mEPP) and evoked endplate potential (eEPP) (*Ling et al., 2010*; *Zanetti et al., 2018*). mEPP, a response that occurs when spontaneously released acetylcholine binds to nicotinic AChR without nerve stimulation, was measured ex vivo near the NMJs of the EDL muscles in the control and *SMN2* 1-copy *Smn1*$^{\Delta MPC}$ mice (*Figure 5A*). The mEPP amplitude was increased in *SMN2* 1-copy *Smn1*$^{\Delta MPC}$ mice (*Figure 5B*), whereas mEPP frequency was comparable between the controls and mutants (*Figure 5C*). The results indicate that the NMJ synapses of *SMN2* 1-copy *Smn1*$^{\Delta MPC}$ mice are functional and more sensitive to acetylcholine compared to the controls. Next, we measured the eEPP by stimulating an action potential at the peroneal nerve (*Figure 5D*). Despite the increased mEPP amplitude, the amplitude of eEPPs was significantly decreased in *SMN2* 1-copy *Smn1*$^{\Delta MPC}$ mice (*Figure 5E*). These results suggest that the nerve terminals in *SMN2* 1-copy *Smn1*$^{\Delta MPC}$ mice exhibit decreased quantal content. This could be due to a decrease in vesicle release probability. Notably, there was no difference in the paired-pulse response, indicating normal neurotransmitter release probability (*Figure 5F, G*). Taken together, these findings suggest that the presynaptic neurotransmission ability of the NMJ is reduced in *SMN2* 1-copy *Smn1*$^{\Delta MPC}$ mutants.

## Disturbed nerve terminal structure in *SMN2* 1-copy *Smn1*$^{\Delta MPC}$ mice

To examine the NMJ ultrastructure of *SMN2* 1-copy *Smn1*$^{\Delta MPC}$ mutants, we utilized transmission electron microscopy (TEM) (*Figure 6A*). The density of junctional folds in *SMN2* 1-copy *Smn1*$^{\Delta MPC}$ mutant specimens was comparable to that of the control (*Figure 6B*). However, the density of synaptic vesicles was substantially elevated in the *SMN2* 1-copy mutants (*Figure 6C*). Since previous electrophysiological results suggested a decrease in presynaptic neurotransmission capacity in *SMN2* 1-copy mutants, this could be due to synaptic vesicles failing to fuse with the membrane, leading to the accumulation of vesicles in the terminal and reduced quantal contents in *Figure 5*. In *Figure 3H*, larger synaptophysin coverage in the mutant may be caused by this synaptic vesicle accumulation. Additionally, the detachment of the nerve terminal is more frequent at the NMJ of mutants (*Figure 6D*). The detachment of nerve terminals observed in *SMN2* 1-copy *Smn1*$^{\Delta MPC}$ mutants could have also resulted in diminished presynaptic neurotransmission capacity. Collectively, these findings indicate that *SMN2* 1-copy *Smn1*$^{\Delta MPC}$ mutants have nerve terminal-specific pathological defects at the NMJ ultrastructural level.

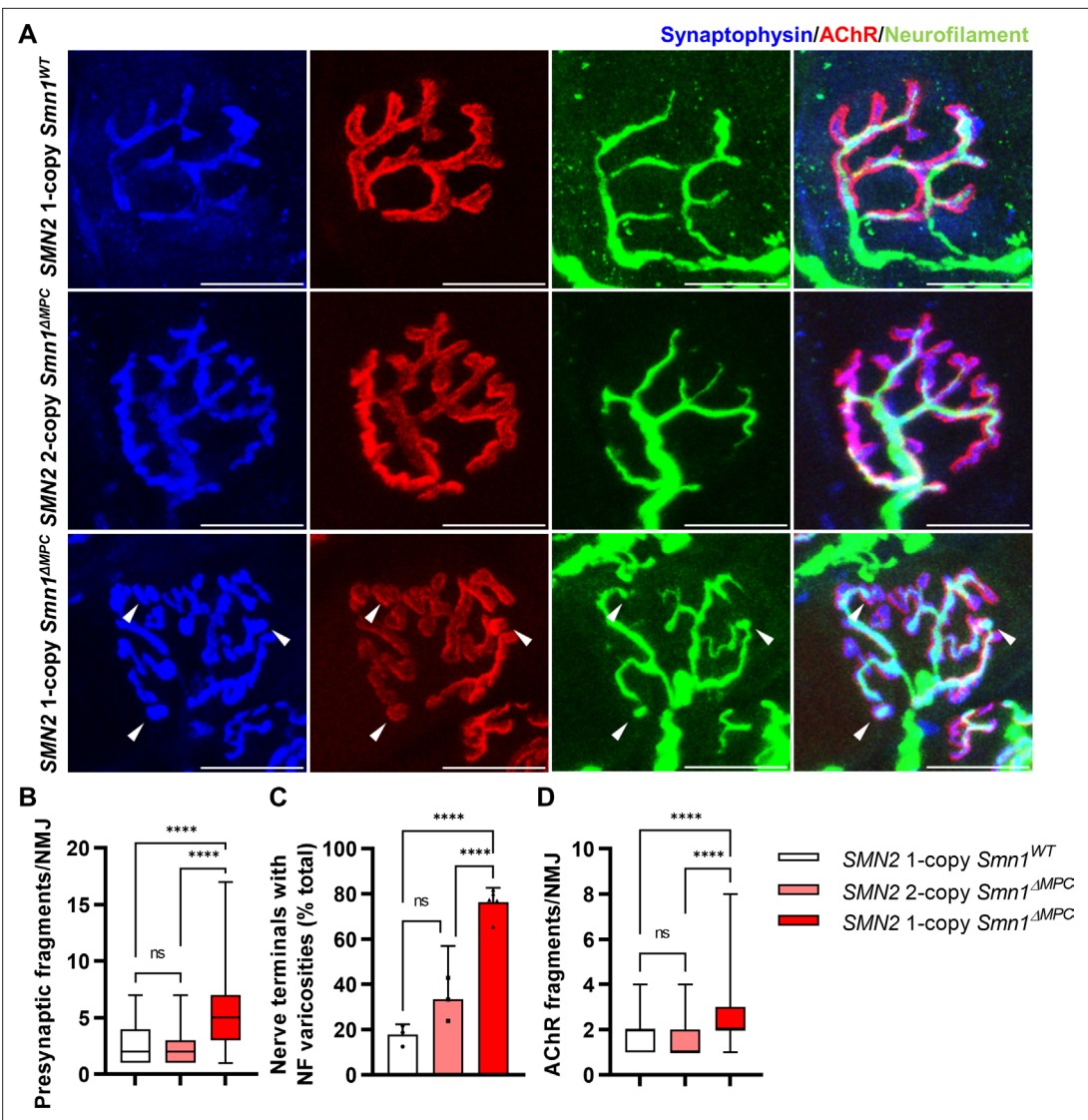

**Figure 4.** Morphological deterioration in neuromuscular junctions (NMJs) of adult *SMN2* 1-copy *Smn1^{ΔMPC}* mice. (**A**) Immunostaining of NMJs in TA muscle of *SMN2* 1-copy *Smn1^{WT}*, *SMN2* 2-copy, and *SMN2* 1-copy *Smn1^{ΔMPC}* mice at P56 with anti-NF (green), anti-synaptophysin (blue), and α-Btx staining acetylcholine receptor (AChR; red). Scale bars, 20 µm. The confocal images of NMJs showed fragmentation and bouton-like neurofilament (NF) varicosities (indicated by arrowheads) in the *SMN2* 1-copy *Smn1^{ΔMPC}* mice. The NMJs of the *SMN2* 1-copy mutant displayed fragmented presynapse (**B**), endplate (**D**), and NF varicosities (**C**) compared to *SMN2* 1-copy *Smn1^{WT}* and *SMN2* 2-copy *Smn1^{ΔMPC}* mice (n = 3–5 mice in each genotype; Presynaptic fragments and AChR fragments: Brown–Forsythe and Welch analysis of variance (ANOVA) with Games–Howell's test; NF varicosities: one-way ANOVA with Tukey's post hoc test). ns; not significantly different. ****p < 0.0001. All box-and-whisker plots show the median, interquartile range, minimum, and maximum. For the box-and-whisker plots, range bars show minimum and maximum (**B, D**). For the bar graph, error bars show standard error of the mean (SEM) (**C**).

## FAPs transplantation rescues NMJ morphology in limb mesenchymal SMN mutants

SMN-deleted limb mesenchymal tissues in *SMN2* 1-copy *Smn1^{ΔMPC}* mutants comprise not only FAPs, but also bone, cartilage, pericytes, and tendon, among others (*Leinroth et al., 2022*; *Nassari et al., 2017*). To evaluate the critical role of FAPs in the postnatal development of the NMJ, we isolated fluorescent protein-labeled wild-type FAPs from *Prrx1^{Cre}*; *Rosa26^{LSL-YFP/+}* or *Prrx1^{Cre}*; *Rosa26^{LSL-tdTomato/+}* mice and transplanted them into the TA muscles of *SMN2* 1-copy *Smn1^{ΔMPC}* mice at postnatal day 10. The TA muscle on the contralateral side was treated with a vehicle as a control. In *SMN2* 1-copy *Smn1^{ΔMPC}* mice at P56, the muscle that received FAPs showed decreased presynaptic fragmentation, NF varicosities, and postsynaptic fragmentation compared to the contralateral muscle (*Figure 7A–D*).

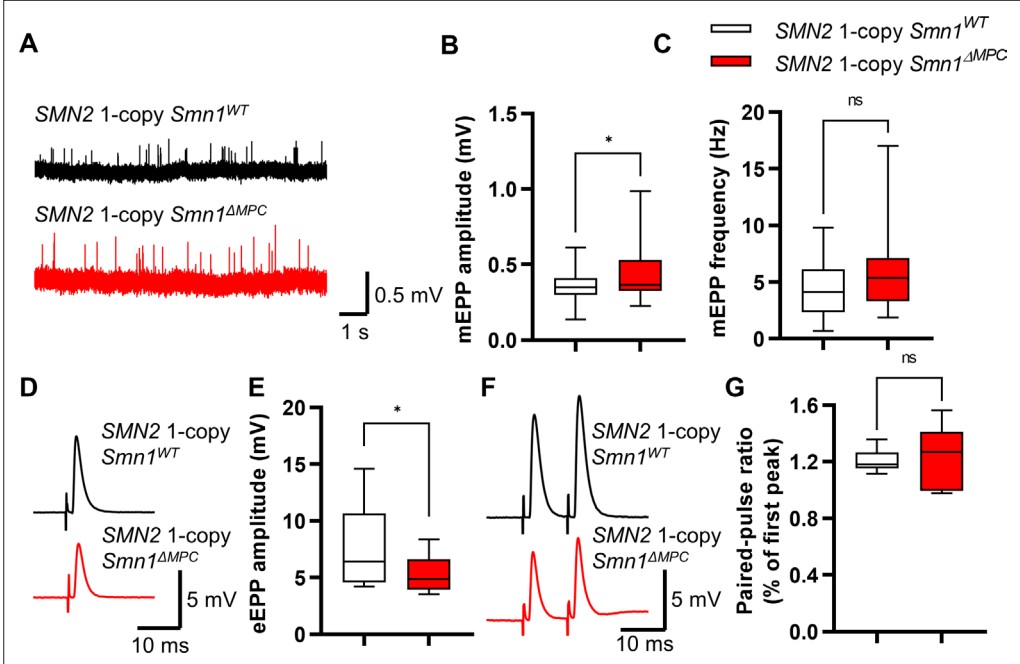

**Figure 5.** Reduced presynaptic neurotransmission ability in the neuromuscular junctions (NMJs) of *SMN2* 1-copy *Smn1^ΔMPC^* mice. (**A**) Representative traces of Miniature endplate potential (mEPP) from *SMN2* 1-copy *Smn1^ΔWT^* (top) and *SMN2* 1-copy *Smn1^ΔMPC^* (bottom) mice. (**B, C**) *SMN2* 1-copy mutant's NMJs showed an increase in mEPP amplitude and no differences in mEPP frequency (1-copy control, n = 25, 9 mice; 1-copy mutant, n = 21, 8 mice; unpaired *t*-test with Welch's correction). (**D**) Representative traces of evoked endplate potential (eEPP) from *SMN2* 1-copy *Smn1^ΔWT^* (top) and *SMN2* 1-copy *Smn1^ΔMPC^* (bottom) mice. (**E**) The mutant's NMJs showed a stronger amplitude of eEPPs (1-copy control, n = 12, 4 mice; 1-copy mutant, n = 12, 3 mice; unpaired *t*-test with Welch's correction). (**F**) Representative traces of paired-pulse response from *SMN2* 1-copy *Smn1^ΔWT^* (top) and *SMN2* 1-copy *Smn1^ΔMPC^* (bottom) mice. (**G**) Paired-pulse response was not different between *SMN2* 1-copy control and mutant NMJs, indicating a comparable neurotransmitter release probability (1-copy control, n = 8, 3 mice; 1-copy mutant, n = 6, 3 mice; unpaired *t*-test with Welch's correction). The electrophysiological recording was performed in the extensor digitorum longus (EDL) muscle at P56. ns: not significantly different. *p < 0.05. All box-and-whisker plots show the median, interquartile range, minimum, and maximum. For the box-and-whisker plots, range bars show minimum and maximum (**B, C, E, G**).

As a result, the transplanted muscle was not significantly different from the control except for NF varicosities. These data demonstrate that the transplantation of wild-type FAPs rescues the abnormal NMJ development in *SMN2* 1-copy *Smn1^ΔMPC^* mice. Overall, our findings indicate that SMN depletion in FAPs leads to the neuronal SMN-independent NMJ pathology in severe SMA, which is rescued by the transplantation of healthy FAPs.

## Discussion

In this paper, we elucidate the contribution of SMN depletion in mesenchymal progenitors for the pathogenesis of SMA. To test this hypothesis, we generated conditional knockout mouse strains to delete the *Smn1* allele specifically in limb mesenchymal cells and carry human *SMN2* copies. Our research using these mouse models resulted in three major discoveries. First, SMN deficiency in FAPs contributes to NMJ pathological defects in SMA. We observed delayed NMJ maturation and varicosities in juvenile *SMN2* 1-copy *Smn1^ΔMPC^* mutant. The pathogenic NMJ phenotypes were also observed in the SMAΔ7 mutant, which is one of the severe SMA mouse models (*Kong et al., 2009*; *Martinez et al., 2012*; *Kariya et al., 2008*). As the SMAΔ7 mutant typically lives for approximately 12 days, the fragmentation of the NMJ in adult *SMN2* 1-copy mutant was not evaluated in the severe SMA mutant. Nevertheless, models that induce a conditional adult SMN deficiency through either Cre^ER^ allele or oligonucleotide administration resulting in SMN reduction, demonstrated the depletion of SMN throughout the body caused fragmentation of NMJ (*Sahashi et al., 2013*; *Kariya et al., 2014*).

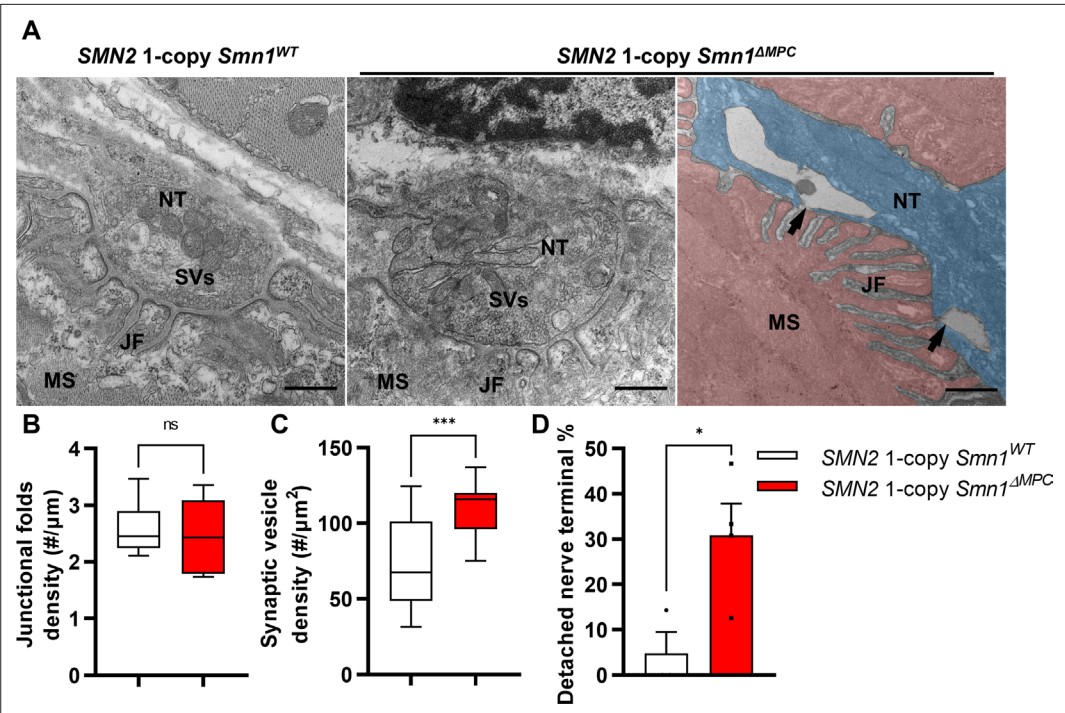

**Figure 6.** Abnormal nerve terminal ultrastructure in *SMN2* 1-copy mutant. (**A**) Representative transmission electron microscopy (TEM) images from neuromuscular junctions (NMJs) of *SMN2* 1-copy *Smn1*^ΔWT and *SMN2* 1-copy *Smn1*^ΔMPC mice at P56. Scale bars, 500 nm. Nerve terminal (NT; indicated by the blue zone). Synaptic vesicles (SVs). Muscle fiber (MS; indicated by the red zone). Endplate junctional folds (JF). Nerve terminal detachment (indicated by arrow) was observed in *SMN2* 1-copy *Smn1*^ΔMPC mice. (**B**) The density of junctional folds in the NMJ of *SMN2* 1-copy *Smn1*^ΔMPC mice no significant change compared to the control, whereas (**C**) the density of synaptic vesicles was increased (*n* = 3–4 mice in each genotype; unpaired *t*-test with Welch's correction). (**D**) The detachment of the nerve terminal occurs more frequently at the NMJ of mutants (*n* = 3–4 mice in each genotype; unpaired *t*-test with Welch's correction). ns: not significantly different. *p < 0.05; ***p < 0.001. All box-and-whisker plots show the median, interquartile range, minimum, and maximum. For the box-and-whisker plots, range bars show minimum and maximum (**B, C**). For the bar graph, error bars show standard error of the mean (SEM) (**D**).

Thus, we demonstrate that the *SMN2* 1-copy *Smn1*^ΔMPC mutant model mimics whole-body SMA mouse models in NMJ morphology. In the electrophysiological test, the SMAΔ7 mutant exhibited reduced quantal content, readily releasable pool, and vesicle release probability (*Torres-Benito et al., 2011*). In our study, we did not directly assess the readily releasable pool in *SMN2* 1-copy *Smn1*^ΔMPC mutant by stimulus train of electrophysiological recording but instead showed reduced quantal content and normal vesicle release probability. In previous studies, it was theorized that the decrease in quantal contents in SMAΔ7 mutants resulted from decreased synaptic vesicle density at nerve terminals caused by motor neuron defects and abnormal axonal transport (*Dale et al., 2011*; *Kong et al., 2009*). However, we found that synaptic vesicle density was increased in the *SMN2* 1-copy *Smn1*^ΔMPC mutant with SMN-sufficient motor neurons. It is possible that the alteration of active zones, which were also altered in the SMA motor terminals (*Kong et al., 2009*), contributed to the reduction in synaptic vesicle fusion and the decreased quantal contents. Indeed, nerve terminal detachment in *SMN2* 1-copy mutant mice was also found in active zone complex protein integrin-α3 knockout mice (*Ross et al., 2017*). Based on the rescue data of transplanted healthy FAPs, we can report that FAP-specific SMN depletion is involved in NMJ pathology of SMA.

Second, we demonstrated that skeletal growth defects, a phenotype observed in SMA (*Khatri et al., 2008*; *Vai et al., 2015*; *Wasserman et al., 2017*; *Baranello et al., 2019*; *Hensel et al., 2020*), are a cell-autonomous pathological effect of the depletion of SMN in bone-forming cells. We observed reduced bone size and volume in juvenile *SMN2* 1-copy *Smn1*^ΔMPC mutant. Depletion of SMN in chondrocytes showed growth plate homeostasis problems with chondrocyte-secreted IGF defects. However, our skeletal study has a limitation in that we did not assess potential impairment of

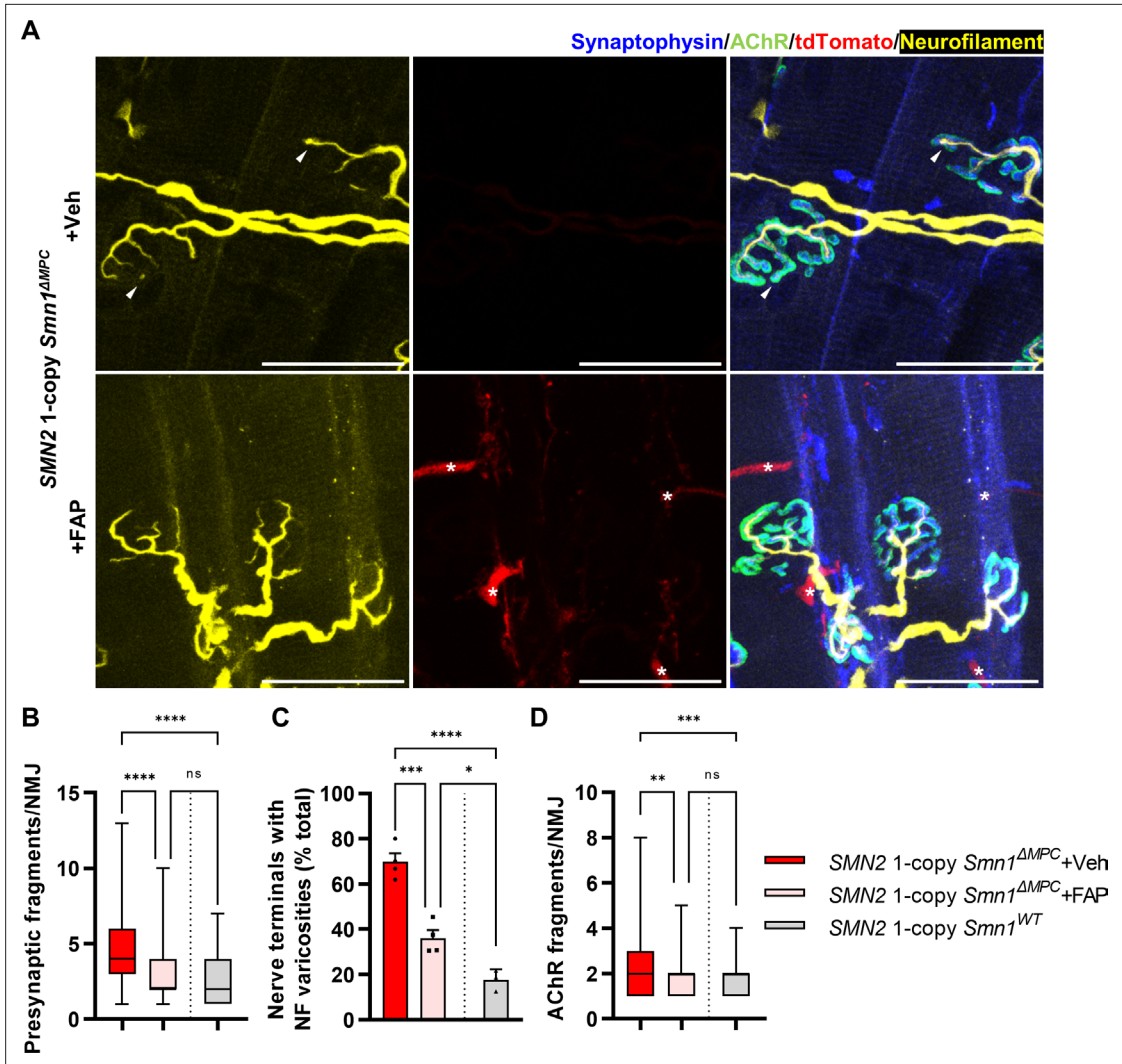

**Figure 7.** Improved postnatal neuromuscular junction (NMJ) development in the TA muscle of *SMN2* 1-copy *Smn1*^ΔMPC mice following healthy fibro-adipogenic progenitors (FAPs) transplantation. (**A**) Immunostaining of NMJs in tdTomato⁺ FAP-transplanted TA muscle (+FAP) and vehicle-treated contralateral muscle (+Veh) in *SMN2* 1-copy *Smn1*^ΔMPC mice at P56 with anti-NF (yellow), anti-synaptophysin (blue), α-Btx staining acetylcholine receptor (AChR; green), and tdTomato fluorescence (red). Scale bars, 40 μm. The images revealed neurofilament (NF) varicosities (indicated by arrowheads) in the +Veh NMJs. The tdTomato⁺ FAPs (marked by asterisks) were transplanted into +FAP NMJs, which exhibited (**B**) decreased presynaptic fragmentation, (**C**) NF varicosities, and (**D**) AChR fragmentation compared to +Veh NMJs and similar to wild-type NMJs (*n* = 3–4 mice in each group; unpaired *t*-test with Welch's correction). ns: not significantly different. *p < 0.05, **p < 0.01, ***p < 0.001, ****p < 0.0001. All box-and-whisker plots show the median, interquartile range, minimum, and maximum. For the box-and-whisker plots, range bars show minimum and maximum (**B, D**). For the bar graph, error bars show standard error of the mean (SEM) (**C**).

endochondral ossification during embryonic development. It may be necessary to further evaluate the embryonic period in addition to the postnatal growth plate defects observed in our study, as cells of the *Prrx1*-lineage are involved in mesenchymal condensation and chondrocyte differentiation during embryonic endochondral ossification (*Hallett et al., 2019*; *Racine and Serrat, 2020*). Previous studies have reported that *IGF1* overexpression improves biochemical and behavioral manifestations in SMA mice, suggesting potential therapies for SMA (*Bosch-Marcé et al., 2011*; *Tsai et al., 2014*). Our study showed that these IGF therapies for SMA could be one to consider for treating bone growth abnormalities. In addition, the skeletal growth defect may affect physiologic muscle atrophy through imbalanced muscle contraction and reduced tension. This physiologic atrophy may be part of SMA-associated muscle weakness independent of neurogenic atrophy.

Third, it was demonstrated that adequate levels of SMN protein are essential for MPCs to contribute to limb neuromusculoskeletal development. The mutants with only 1 copy of *SMN2* exhibit problematic

symptoms observed in both SMA patients and mouse models, while *SMN2* 2-copy mutants display a typical phenotype in bone and NMJ. The lack of SMN protein in MPCs by insufficient *SMN2* copies, similar to the deficiency seen in severe SMA, is responsible for the onset of SMA pathology. Thus, we propose that restoration of deficient SMN in MPCs is crucial for rehabilitating their function. Based on these discoveries, SMN replenishment treatments for MPCs, specifically FAPs, and chondrocytes, are necessary to provide a complete solution for neuromusculoskeletal defects in severe SMA patients.

The initial focus for the treatment of SMA was on motor neurons located in the spinal cord, and pharmaceuticals were created to address the deficiency of SMN in these neurons (*Mercuri et al., 2020*). For example, intrathecal injections of Spinraza can efficiently boost SMN in the CNS, including the spinal cord (*Passini et al., 2011*; *Claborn et al., 2019*). However, Spinraza does not address the lack of SMN in peripheral tissues, including mesenchymal cells, highlighted in recent studies. The drug Zolgensma, which employs AAV9 to express SMN through systemic delivery, appears to resolve this issue (*Foust et al., 2009*; *Valori et al., 2010*; *Mattar et al., 2013*). Nevertheless, a prior study indicates that chondrocytes within the growth plate and articular cartilage do not get infected with AAV9 (*Yang et al., 2019*). Furthermore, it is well established that DNA vectors delivered via AAV9 undergo dilution during cell proliferation (*Penaud-Budloo et al., 2008*; *Colella et al., 2018*; *Van Alstyne et al., 2021*; *Heller et al., 2021*). Given the infection issue and the dilution issue by the active cell population changes of chondrocytes and FAPs during postnatal development (*Petrany et al., 2020*; *Bachman and Chakkalakal, 2022*), SMN supplementation through Zolgensma alone would not be sufficient for severe SMA patients. Thus, there is an urgent need to research and develop therapeutic strategies that target mesenchymal progenitors.

While this study is the first to demonstrate the impact of SMN depletion in FAPs on the NMJ development, it does not elucidate the specific mechanism by which FAPs influence the NMJ. Our study observed NMJ recovery in the muscles transplanted with healthy FAPs, but not in the contralateral muscles, indicating that FAPs are likely involved in NMJ development through juxtacrine or paracrine signaling. Since FAPs interact with surrounding tissues through a variety of signaling factors, such as extracellular matrix (*Contreras et al., 2021*; *Scott et al., 2019*), Wnt-related protein (*Lukjanenko et al., 2019*), and Bmp signaling protein (*Uezumi et al., 2021*; *Camps et al., 2020*), mis-splicing of the signaling factors due to SMN reduction could disturb the homeostasis of neighboring tissues (*Zhang et al., 2008*). Furthermore, the *Hsd11b1*-positive subpopulation of FAPs associated with NMJ was discovered through single-cell RNA sequencing in a recent study (*Leinroth et al., 2022*). This population is located adjacent to the NMJ and responds to denervation, indicating an increased possibility of interaction with the NMJ organization. Therefore, it is necessary to conduct additional investigations into the expression of various signaling factors by diverse FAP subpopulations in future studies.

## Methods

### Key resources table

| Reagent type (species) or resource | Designation | Source or reference | Identifiers | Additional information |
|---|---|---|---|---|
| Genetic reagent (*Mus musculus*) | *Prrx1^Cre^* | The Jackson Laboratory | Strain #: 005584; RRID:IMSR_JAX:005584 | |
| Genetic reagent (*M. musculus*) | *Smn1^f7/+^* | The Jackson Laboratory | Strain #: 006138; RRID:IMSR_JAX:006138 | |
| Genetic reagent (*M. musculus*) | *Rosa26^LSL-YFP/+^* | The Jackson Laboratory | Strain #: 006148; RRID:IMSR_JAX:006148 | |
| Genetic reagent (*M. musculus*) | *Rosa26^LSL-tdTomato/+^* | The Jackson Laboratory | Strain #: 007914; RRID:IMSR_JAX:007914 | |
| Genetic reagent (*M. musculus*) | *SMN2^+/+^; Smn1^+/−^* | The Jackson Laboratory | Strain #: 005024; RRID:IMSR_JAX:005024 | |
| Genetic reagent (*M. musculus*) | *Smn1^+/−^; SMN2^+/+^; SMNΔ7^+/+^* | The Jackson Laboratory | Strain #: 005025; RRID:IMSR_JAX:005025 | |
| Antibody | anti-SMN (Mouse monoclonal) | BD Biosciences | Cat. #: 610646; RRID:AB_397973 | WB (1:1000) |

*Continued on next page*

*Continued*

| Reagent type (species) or resource | Designation | Source or reference | Identifiers | Additional information |
|---|---|---|---|---|
| Antibody | anti-alpha-tubulin (Rabbit monoclonal) | Abcam | Cat. #: ab176560; RRID:AB_2860019 | WB (1:1000) |
| Antibody | anti-Neurofilament M (Rabbit polyclonal) | Merckmillipore | Cat. #: ab1987; RRID:AB_91201 | IF (1:1000) |
| Antibody | anti-Synaptophysin 1 (Guinea pig polyclonal) | Synaptic Systems | Cat. #: 101 004; RRID:AB_1210382 | IF (1:500) |
| Antibody | anti-GFP (Chicken polyclonal) | Abcam | Cat. #: ab13970; RRID:AB_300798 | IF (1:500) |
| Antibody | anti-Ki67 (Rabbit polyclonal) | Abcam | Cat. #: ab15580; RRID:AB_443209 | IF (1:500) |
| Antibody | anti-Itgb3 (Rabbit monoclonal) | Cell Signaling | Cat. #: 13166; RRID:AB_2798136 | IF (1:100) |
| Antibody | anti-p-AKT(S473) (Rabbit monoclonal) | Cell Signaling | Cat. #: 4060; RRID:AB_2315049 | IF (1:100) |
| Antibody | anti-CD45-APC (Rat monoclonal) | Biolegend | Cat. #: 103111; RRID:AB_312976 | FACS (3 µl per test) |
| Antibody | anti-CD31-APC (Rat monoclonal) | Biolegend | Cat. #: 102409; RRID:AB_312904 | FACS (3 µl per test) |
| Antibody | anti-Sca-1(Ly6a)-FITC (Rat monoclonal) | Biolegend | Cat. #: 122507; RRID:AB_756192 | FACS (3 µl per test) |
| Antibody | anti-Sca-1(Ly6a)-Pacific blue (Rat monoclonal) | Biolegend | Cat. #: 108120; RRID:AB_493273 | FACS (3 µl per test) |
| Antibody | anti-Vcam1-Biotin (Rat monoclonal) | Biolegend | Cat. #: 105703; RRID:AB_313204 | FACS (3 µl per test) |
| Antibody | anti-Rabbit IgG-HRP (Goat monoclonal) | Promega | Cat. #: W4011; RRID:AB_430833 | WB (1:10,000) |
| Antibody | anti-Mouse IgG-HRP (Goat monoclonal) | Promega | Cat. #: W4021; RRID:AB_430834 | WB (1:10,000) |
| Antibody | anti-Rabbit IgG-Alexa fluor 488 (Goat monoclonal) | Invitrogen | Cat. #: A11034; RRID:AB_2576217 | IF (1:500) |
| Antibody | anti-Chicken IgY-Alexa fluor 488 (Goat monoclonal) | Invitrogen | Cat. #: A11039; RRID:AB_2534096 | IF (1:500) |
| Antibody | anti-Rabbit IgG-Alexa fluor Plus 647 (Goat monoclonal) | Invitrogen | Cat. #: A32733; RRID:AB_2633282 | IF (1:500) |
| Antibody | anti-Guine pig IgG-Alexa fluor 405 (Goat monoclonal) | Abcam | Cat. #: ab175678; RRID:AB_2827755 | IF (1:500) |
| Peptide, recombinant protein | PE-Cy7-Streptavidin | Biolegend | Cat. #: 405206 | FACS (3 µl per test) |
| Peptide, recombinant protein | Alpha-bungarotoxin-Alexa fluor 555 | Invitrogen | Cat. #: B35451 | IF (1:1000) |
| Peptide, recombinant protein | Alpha-bungarotoxin-Alexa fluor 488 | Invitrogen | Cat. #: B13422 | IF (1:1000) |
| Peptide, recombinant protein | µ-conotoxin GIIIB | Alomone | Cat. #: C-270 | Electrophysiology (2.5 µM) |
| Commercial assay or kit | Pierce BCA protein assay kits | Thermo Fisher Scientific | Cat. #: C-23225 | |
| Software | ImageJ | ImageJ | RRID:SCR_003070 | |
| Software | AccuCT | PerkinElmer | | |
| Software | Microsoft Excel | Microsoft | RRID:SCR_016137 | |
| Software | Leica Application Suite X | Leica | RRID:SCR_013673 | |
| Software | GraphPad Prism | Graphpad | RRID:SCR_002798 | |
| Software | pClamp | Molecular devices | RRID:SCR_011323 | |

*Continued on next page*

*Continued*

| Reagent type (species) or resource | Designation | Source or reference | Identifiers | Additional information |
|---|---|---|---|---|
| Software | QIAGEN | QIAGEN | RRID:SCR_008539 | |
| Other | Gill No. 3 formula hematoxylin | Sigma | Cat. #: GHS332 | Hematoxylin staining solution (1×) |
| Other | Eosin Y solution | Sigma | Cat. #: HT110116 | Eosin staining solution (1×) |
| Other | Toluidine blue | Sigma | Cat. #: 198161 | Toluidine blue staining (1%) |

## Animals

*Prrx1$^{Cre}$* (stock 005584), *Smn1$^{f7/+}$* (stock 006138), *Rosa26* $^{LSL-YFP/+}$ (stock 006148), *Rosa26* $^{LSL-tdTomato/+}$ (stock 007914), *SMN2$^{+/+}$*, *Smn1$^{+/-}$* (stock 005024 – *Smn1* knockout and *SMN2* homologous transgenic mouse) and *Smn1$^{+/-}$*; *SMN2$^{+/+}$*; *SMNΔ7$^{+/+}$* (stock 005025 005024 – *Smn1* knockout and *SMN2*, *SMNΔ7* homologous transgenic mouse) mice were acquired from The Jackson Laboratory (Bar Harbor, ME, USA). *SMN2* 0-copy *Smn1$^{ΔMPC}$* mice (*Prrx1$^{Cre}$*; *Smn1$^{f7/f7}$*), *SMN2* 1-copy *Smn1$^{ΔMPC}$* mice (*Prrx1$^{Cre}$*; *Smn1$^{f7/f7}$*; *SMN2$^{+/0}$* – *SMN2* heterologous allele), and *SMN2* 2-copy *Smn1$^{ΔMPC}$* mice (*Prrx1$^{Cre}$*; *Smn1$^{f7/f7}$*; *SMN2$^{+/+}$* – *SMN2* homologous allele) were generated by crossing *Prrx1$^{Cre}$* mice with *Smn1$^{f7/+}$* and *Smn1$^{+/-}$*; *SMN2$^{+/+}$* mice. To utilize FAPs transplantation, we generated *Prrx1$^{Cre}$*; *Rosa26* $^{LSL-YFP/+}$ mice and *Prrx1$^{Cre}$*; *Rosa26* $^{LSL-tdTomato/+}$ mice by breeding *Prrx1$^{Cre}$* with *Rosa26* $^{LSL-YFP/+}$ and *Rosa26* $^{LSL-tdTomato/+}$ mice, respectively. To avoid deletion of the floxed allele by *Prrx1$^{Cre}$* expression in female germline, we bred females without *Prrx1$^{Cre}$* line to *Prrx1$^{Cre}$* transgenic males. SMAΔ7 mutants (*Smn1$^{-/-}$*; *SMN2$^{+/+}$*; *SMNΔ7$^{+/+}$*) were produced by mating *Smn1$^{+/-}$*; *SMN2$^{+/+}$*; *SMNΔ7$^{+/+}$* mice. Both male and female mice were used in the experiments, and no sex-specific differences were observed. Control littermates lacking *Prrx1$^{Cre}$* were utilized for analysis. All mouse lines were housed under controlled conditions with specific pathogen free and handled according to the guidelines of the Seoul National University Institutional Animal Care and Use Committee (Protocol number: SNU-210313-1).

## Micro-CT

The femurs from three groups – control, *SMN2* 2-copy *Smn1$^{ΔMPC}$*, and *SMN2* 1-copy *Smn1$^{ΔMPC}$* mice at postnatal day 14 (P14) – were isolated and cleaned of muscles and skin. Subsequently, the femurs were preserved in 4% paraformaldehyde (PFA) in phosphate-buffered saline (PBS) overnight at 4°C before micro-CT. The femurs were then imaged through a Quantum GX II micro-CT imaging system (PerkinElmer, Hopkinton, MA, USA). The X-ray source for scanning was set at 90 kV and 88 mA with a field of view of 10 mm (voxel size, 20 µm; scanning time, 14 min). The 3D imaging was viewed using the 3D Viewer software of the Quantum GX II. The size and volume of the femur bone were measured via AccuCT analysis software within the ossified diaphysis and metaphysis of the femur, excluding the epiphysis. BMD calibration was performed using a 4.5-mm BMD phantom and BMD measurements were taken at the center of the diaphysis.

## Histology

Alcian blue and alizarin red staining was performed on control and *SMN2* 0-copy *Smn1$^{ΔMPC}$* mice at E18.5 using a previously reported protocol (*Ovchinnikov, 2009*). For H&E and toluidine blue staining of the growth plate in the control and *SMN2* 1-copy *Smn1$^{ΔMPC}$* mice at P14, the femurs were fixed in 4% PFA for 24 hr, rinsed in running tap water for 24 hr, and incubated with 10% ethylenediaminetetraacetic acid (EDTA) (pH 7.4) at 4°C with shaking for 2–3 days. Subsequently, the samples were rinsed in running tap water for 24 hr and then dehydrated through ethanol/xylene and embedded in paraffin. The embedded samples were then sectioned to a thickness of 5 µm, rehydrated, and stained with Gill No. 3 formula hematoxylin and eosin Y (H&E, Sigma-Aldrich, St. Louis, MO, USA) and toluidine blue (Sigma-Aldrich, St. Louis, MO, USA). Stained slides were analyzed using ×10 and ×20 objectives in the EVOS M7000 imaging system (Thermo Fisher Scientific, Waltham, MA, USA). The growth plate's proliferative and hypertrophic zones were defined by their respective cell sizes.

## Hindlimb FAPs isolation

Isolation of limb muscle FAPs was performed according to a previously reported protocol (*Kim et al., 2020*) with modifications. Limb muscles were dissected and mechanically dissociated in Dulbecco's

modified Eagle's medium (DMEM, Hyclone) containing 10% horse serum (Hyclone, Logan, UT, USA), collagenase II (800 units/ml; Worthington, Lakewood, NJ, USA), and dispase (1.1 units/ml; Thermo Fisher Scientific, Waltham, MA, USA) at 37°C for 40 min. Digested suspensions were subsequently triturated by sterilized syringes with 20 G 1/2 needle (BD Biosciences, Franklin Lakes, NJ, USA) and washed with DMEM to harvest mononuclear cells. Mononuclear cells were stained with corresponding antibodies. All antibodies used in fluorescence-activated cell sorting (FACS) analysis are listed in *Supplementary file 1*. To exclude dead cells, 7AAD (Sigma-Aldrich; St. Louis, MO, USA) was used. Stained cells were analyzed and 7AAD⁻Lin⁻Vcam⁻Sca1⁺ (stem cell antigen 1; *Ly6a*) (FAPs) were isolated using FACS Aria III cell sorter (BD Biosciences) with four-way purity precision. For western blot, freshly isolated FAPs were cultured at 37°C in alpha-MEM (Hyclone) supplemented with Antibiotic–Antimycotic (anti–anti, Gibco) and 20% fetal bovine serum (FBS; Hyclone). For transplantation, YFP⁺ FAPs and tdTomato⁺ FAPs were isolated from postnatal day 10–21 *Prrx1^Cre*; *Rosa26^{LSL-YFP/+}* mice and *Prrx1^Cre*; *Rosa26^{LSL-tdTomato/+}* mice, respectively. Isotype control density plots were used as a reference for positive gating.

## Chondrocytes isolation

Isolation of chondrocytes followed a modified protocol previously published (*Jonason et al., 2015*). Using blunt forceps, cartilage caps were removed from P21 mouse femoral heads and dissected into ~1 mm fragments in a Petri dish with 10× anti–anti in PBS. The cartilage fragments were washed twice with PBS and then incubated in 5 ml collagenase II solution (800 units/mL collagenase II in DMEM with anti–anti, sterilized by 0.2 μm filtration) in a 60 mm culture dish at 37°C in a 5% CO₂ incubator overnight. Chondrocytes were released by pipetting the remaining cartilage fragments 10 times and then filtering them through a 70-μm cell strainer to a 50-ml conical tube. The cells were then washed twice with PBS and pelleted by centrifugation at 500 × *g* for 5 min. Subsequently, the cells were cultured overnight at 37°C in a 5% CO₂ incubator, in complete culture medium (DMEM with 10% FBS and anti–anti) overnight at 37°C in a 5% CO₂ incubator.

## PCR reaction

To detect the presence of *Smn1* exon 7 floxed and deleted allele, 20 mg of each tissue were dissolved in direct PCR buffer (VIAGEN) with proteinase K overnight at 65°C. After inactivation at 95°C for 30 min, PCR was performed using previously reported primer sets to verify the existence of *Cre*, *Smn1^{f7}*, and *Smn1^{Δ7}* alleles (*Frugier et al., 2000*). Primers are listed in .

## RNA extraction and measurement of mRNA expression

Total RNA was extracted from the brain, liver, spinal cord L4, tibialis anterior muscle, freshly isolated FAPs, and chondrocytes using a TRIzol Reagent (Life Technologies, Carlsbad, CA, USA) and analyzed by qRT-PCR. First-strand complementary DNA was synthesized from 1 μg of RNA using ReverTra Ace (Toyobo, Osaka, Japan) containing random oligomer according to the manufacturer's instructions. qRT-PCR (QIAGEN) was performed with SYBR Green technology (SYBR Premix Ex Taq, QIAGEN) using specific primers against indicated genes. Relative mRNA levels were determined using the $2^{-\Delta\Delta Ct}$ method and normalized to *Gapdh* (*Figure 1H*, *Figure 1—figure supplement 1K*). Primers are listed in *Supplementary file 1*.

## Western blot

Cultured FAPs at passage 3 were homogenized in radioimmunoprecipitation assay (RIPA) buffer (50 mM Tris–HCl, pH 7.5, 0.5% sodium dodecyl sulfate, 20 μg/ml aprotinin, 20 μg/ml leupeptin, 10 μg/ml phenylmethylsulfonyl fluoride, 1 mM sodium orthovanadate, 10 mM sodium pyrophosphate, 10 mM sodium fluoride, and 1 mM dithiothreitol). Cell lysates were centrifuged at 13,000 rpm for 15 min. Supernatants were collected and subjected to immunoblot. BCA protein assay (Thermo Fisher Scientific) was used for estimating total protein concentrations. Normalized total proteins were analyzed by electrophoresis in 10% polyacrylamide gels and transferred to polyvinylidene fluoride (PVDF) membranes (Millipore, Billerica, MA, USA). Membranes were blocked in 5% skim milk (BD Biosciences) in tris-buffered saline (TBS) with 0.1% Tween-20 and incubated with primary antibodies overnight at 4°C. After incubation with the corresponding horseradish peroxidase (HRP)-conjugated secondary antibodies, the membranes were developed using a Fusion solo chemiluminescence imaging system

(Vilber, Marne-la-Vallée, France). α-Tubulin was used as a loading control. Band intensities were quantified using ImageJ software. Antibodies used in this study are listed in *Supplementary file 1*. Primary and secondary antibodies were diluted 1:1000 and 1:10,000 with PBS containing 0.1% Tween-20 and 3% bovine serum albumin, respectively.

## FAPs transplantation

FAPs transplantation was performed according to a previously reported protocol (*Kim et al., 2020*) with modifications. YFP$^+$ or tdTomato$^+$ FAPs (7AAD$^-$Lin$^-$Vcam$^-$Sca1$^+$) were isolated by FACS from the limb muscles of indicated mice. $1 \times 10^5$ FAPs were suspended in 0.1% gelatin (Sigma-Aldrich, St. Louis, MO, USA) in PBS and then transplanted into one side of the TA muscles of *SMN2* 1-copy *Smn1$^{\Delta MPC}$* mice. The contralateral muscle received an equivalent volume of 0.1% gelatin in PBS (Vehicle).

## Electrophysiology

The EDL muscle was dissected from control and *SMN2* 1-copy *Smn1$^{\Delta MPC}$* mice, along with the peroneal nerve, and then pinned to a Sylgard-coated recording chamber. Intracellular recording was conducted in oxygenated Ringer's solution, which comprised 138.8 mM NaCl, 4 mM KCl, 12 mM NaHCO$_3$, 1 mM KH$_2$PO$_4$, 1 mM MgCl$_2$, and 2 mM CaCl$_2$ with a pH of 7.4. Action potential of the muscle was prevented by preparing the muscle in 2.5 µM µ-conotoxin GIIIB (Alomone, Jerusalem, Israel) for 10 min beforehand. The recording was performed in toxin-free Ringer's solution. mEPPs were recorded from a junction, followed by recordings of eEPPs by stimulating the attached peroneal nerve. The eEPPs were elicited using evoked stimulation. Paired-pulse stimulation (10-ms interstimulus interval) was utilized to assess synaptic transmission. The data were obtained and analyzed with Axoclamp 900A and Clampfit version 10.7 software.

## Transmission electron microscopy

NMJ TEM followed a modified protocol previously reported (*Modla et al., 2010*). Mouse EDL muscle was swiftly excised and fixed in 4% PFA dissolved in Sorensen's phosphate buffer (0.1 M, pH 7.2), followed by washing in 0.1 M phosphate buffer. The EDL was then gradually infiltrated on a rotator at room temperature with sucrose: 0.1 M phosphate buffer solutions of 30% and 50%, for 1 hr each, followed by an overnight incubation in 70% sucrose. Excess sucrose was then eliminated using filter paper, and the muscle was embedded in an optimal cutting temperature compound (O.C.T.; Sakura Fineteck, Torrance, CA, USA), followed by being frozen in a cryostat (Leica, Wetzlar, Germany). 10-µm-thick longitudinal sections were washed in PBS and treated with Alexa fluor 555-conjugated α-bungarotoxin (1:500, Invitrogen) for an hour. Imaging was conducted with the EVOS M7000 imaging system, and we selected four to five NMJ-rich regions for processing with TEM. The sections were fixed with 2% glutaraldehyde and 2% PFA in 0.1 M cacodylate buffer (pH 7.2) for 2 hr at room temperature, with an additional overnight incubation at 4°C. After washing with 0.1 M cacodylate buffer they were post-fixed with 1% osmium tetraoxide in 0.1 M cacodylate buffer (pH 7.2) for 2 hr at 4°C. The sections were then stained en bloc with 0.5% uranyl acetate overnight, washed with distilled water, and dehydrated using serial ethanol and propylene oxide. The sections were embedded in epoxy resin (Embed-812, Electron Microscopy Sciences) and detached from the slides by dipping them in liquid nitrogen. Ultra-thin sections (70 nm) were prepared with a diamond knife on an ultramicrotome (ULTRACUT UC7, Leica) and mounted on 100 mesh copper grids. Sections were stained with 2% uranyl acetate for 10 min and lead citrate for 3 min, then observed using a transmission electron microscope (80 kV, JEM1010, JEOL or 120 kV, Talos L120C, FEI). Synaptic vesicle density was quantified within a distance of 500 nm from the presynaptic membrane.

## Immunohistochemistry

For NMJ staining, freshly dissected TA muscles were fixed in 4% PFA for 30 min at room temperature. Subsequently, the muscles were cryoprotected in 30% sucrose overnight, embedded in O.C.T., snap-frozen in liquid nitrogen, and stored at −80°C prior to sectioning. Longitudinal 40-µm-thick sections were obtained from the embedded muscles using a cryostat. The sections were blocked for 2 hr at room temperature using 5% goat serum and 5% bovine serum albumin in PBS/0.4% Triton X-100. Then, the sections were incubated with primary antibodies in the blocking buffer for 2 days at 4°C. After washing the sections three times with PBS/0.4% Triton X-100, the sections were stained with

secondary antibodies overnight at 4°C, and then incubated with Alexa fluor 488- or 555-conjugated α-bungarotoxin (1:500, Invitrogen) for 2 hr at room temperature (RT), washed three times with PBS/0.4% Triton X-100 and mounted in VECTASHIELD. Z-serial images were collected at ×40 with a Leica SP8 confocal laser scanning microscope. To analyze NMJ morphology, LasX software was used to obtain maximal projections. NF varicosity refers to the varicose NF end connected to the rest of the nerve terminal. To quantify NMJ size and synaptophysin coverage, the Btx area and synaptophysin area were measured by ImageJ analysis software.

For bone section staining, the 5-μm-thick bone sections were rehydrated and antigen retrieval was then performed in citrate buffer (10 mM citric acid, pH 6) at 95°C 20 min. The sections were blocked for 1 hr at room temperature using 5% goat serum and 5% bovine serum albumin in PBS/0.4% Triton X-100. Then, the sections were incubated with primary antibodies in the blocking buffer at 4°C overnight. After washing the sections three times with PBS/0.1% Triton X-100, the sections were stained with secondary antibodies for 1 hr at RT, washed and mounted. Imaging was conducted with the EVOS M7000 imaging system.

## Statistical analysis

Sample size determination was based on anticipated variability and effect size that was observed in the investigator's lab for similar experiments. For quantification, individual performing the counts were blinded to sample identity and randomized. All statistical analyses were performed using GraphPad Prism 9 (GraphPad Software). For comparison of significant differences in multiple groups for normally distributed data, statistical analysis was performed by one- or two-way analysis of variance (ANOVA) followed by Tukey's pairwise comparison post hoc test. For non-normally distributed data, Brown–Forsythe and Welch ANOVA followed by Games–Howell multiple comparisons test was used. For the comparison of two groups, Student's unpaired $t$-test assuming a two-tailed distribution with Welch's correction was used. Unless otherwise noted, all error bars represent standard error of the mean. The number of biological replicates and statistical analyses for each experiment were indicated in the figure legends. Independent experiments were performed at least in triplicates. $p < 0.05$ was considered statistically significant at the 95% confidence level. *$p < 0.05$, **$p < 0.01$, ***$p < 0.001$, ****$p < 0.0001$.

## Acknowledgements

We express our gratitude to the Kong and Choi laboratory members for their valuable feedback during the project. This work was supported by the National Research Foundation of Korea (NRF-2022R1A2C3007621, NRF-2020R1A5A1018081, and NRF-2020R1A2C3011464).

## Additional information

### Funding

| Funder | Grant reference number | Author |
| --- | --- | --- |
| National Research Foundation of Korea | NRF-2022R1A2C3007621 | Young-Yun Kong |
| National Research Foundation of Korea | NRF-2020R1A5A1018081 | Young-Yun Kong |
| National Research Foundation of Korea | NRF-2020R1A2C3011464 | Se-Young Choi |

The funders had no role in study design, data collection, and interpretation, or the decision to submit the work for publication.

### Author contributions

Sang-Hyeon Hann, Conceptualization, Data curation, Formal analysis, Validation, Investigation, Visualization, Methodology, Writing – original draft, Writing – review and editing; Seon-Yong Kim, Data curation, Formal analysis, Validation, Investigation, Methodology, Writing – review and editing; Ye Lynne

Kim, Validation, Investigation, Writing – review and editing; Young-Woo Jo, Data curation, Formal analysis, Methodology, Writing – review and editing; Jong-Seol Kang, Formal analysis, Investigation, Methodology; Hyerim Park, Investigation, Visualization, Methodology; Se-Young Choi, Young-Yun Kong, Supervision, Funding acquisition, Methodology, Project administration, Writing – review and editing

**Author ORCIDs**
Sang-Hyeon Hann ⓘ http://orcid.org/0000-0002-0871-3414
Se-Young Choi ⓘ https://orcid.org/0000-0001-7534-5167
Young-Yun Kong ⓘ https://orcid.org/0000-0001-7335-3729

**Ethics**
The care and treatment of animals in this study were approved by the Institutional Animal Care and Use Committee (IACUC) protocols (SNU-210313-1) of Seoul National University.

Reviewer #1 (Public Review): https://doi.org/10.7554/eLife.92731.3.sa1
Reviewer #2 (Public Review): https://doi.org/10.7554/eLife.92731.3.sa2
Reviewer #3 (Public Review): https://doi.org/10.7554/eLife.92731.3.sa3
Author Response https://doi.org/10.7554/eLife.92731.3.sa4

---

# Additional files

### Supplementary files
- MDAR checklist
- Supplementary file 1. A primer list of genomic PCR and qRT-PCR.

### Data availability
All data generated or analyzed during this study are included in the manuscript and supporting files; source data files have been provided for Figure 2.

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
