## [Editor Report · eLife assessment]

This **important** work by Hann et al. advances our understanding of the role of the survival motor neuron (SMN) protein in coordinating pathogenesis of the spinal muscular atrophy (SMA). The authors addressed many concerns raised by the reviewers, providing **convincing** evidence in terms of skeletal analyses not being able to satisfactorily elucidate SMN regulation of bone development.

---

## [Referee Report · Reviewer #1 (Public Review)]

Summary:

The manuscript by Hann et. al examines the role of survival motor neuron protein (SMN) in lateral plate mesoderm-derived cells using the Prrx1Cre to elucidate how changing cell-specific SMN levels coordinate aspects of the spinal muscular atrophy (SMA) pathology. SMN has generally been studied in neuronal cells, and this is one of the first insights into non-neuronal cells that may contribute to SMA disease. The authors generated 3 mouse lines: a Prrx1;Smnf/f conditional null mouse, as well as, single and double copy Prrx1;Smnf/f;SMN2 mice carrying either one or two copies of a human SMN2 transgene. First, the bone development and growth of all three were assessed; the conditional null Smn mutation was lethal shortly after birth, while the SMN2 2-copy mutant did not exhibit bone growth phenotypes. Meanwhile, single-copy SMN2 mutant mice showed reduced size and shorter limbs with shorter proliferative and hypertrophic chondrocyte zones. The authors suggested that this was cell autonomous by assessing the expression of extrinsic factors known to modulate proliferation/differentiation of growth plate chondrocytes. After assessing bone phenotypes, the authors transitioned to the assessments of neuromuscular junction (NMJ) phenotypes, since there are documented neuromuscular impairments in SMA and the Prrx1Cre transgene is expressed in muscle-associated fibro-adipogenic progenitors (FAPs). Neonatal NMJ development was unchanged in mutant mice with two copies of SMN2 , but adult single-copy SMN2 mutant mice had abnormal NMJ morphology, altered presynaptic neurotransmission, and problematic nerve terminal structure. Finally, the authors sought to assess the ability to rescue NMJ phenotypes via FAP cell transplantation and showed wild-type FAPs were able to reduce pre/postsynaptic fragmentation and neurofilament varicosities.

Strengths:

The conditional genetic approaches are novel and interestingly demonstrate the potential for chondrocyte and fibro-adipogenic progenitor-specific contributions to the SMA pathology.

The characterizations of the neuromuscular and NMJ phenotypes are relatively strong.

The data strongly suggest a non-neuronal contribution to SMA, which indicates a need for further mechanistic (cellular and molecular) studies to better understand SMA.

Weaknesses:

The skeletal analyses are not rigorous and likely do not get to the core of how SMN regulates bone development.

The overall work is descriptive and lacks convincing mechanisms.

Additional experimentation is likely needed to fully justify the conclusions.

---

## [Referee Report · Reviewer #2 (Public Review)]

Summary:

Sang-Hyeon et al. laid out a compelling rationale to explore the role of the SMN protein in mesenchymal cells, to determine whether SMN deficiency there could be a pathologic mechanism of SMA. They crossed Smnf7/f7 mice with Prrx1Cre mice to produce SmnΔMPC mice where exon 7 was specifically deleted and thus SMN protein was eliminated in limb mesenchymal progenitor cells (MPCs). To demonstrate gene dosage-dependence of phenotypes, SmnΔMPC mice were crossed with transgenic mice expressing human SMN2 to produce SmnΔMPC mice with different copies of SMN2 (0, 1, or 2). The paper provides genetic evidence that SMN in mesenchymal cells regulates the development of bones and neuromuscular junctions. Genetic data were convincing and revealed novel functions of SMN.

Strengths:

Overall, the paper provided genetic evidence that SMN deficiency in mesenchymal cells caused abnormalities in bones and NMJs, revealing novel functions of SMN and leading to future experiments. As far as genetics is concerned, the data were convincing (except for the rescue experiment, see below); the conclusions are important.

Weaknesses:

The paper seemed to be descriptive in nature and could be improved with more experiments to investigate underlying mechanisms. In addition, some data appeared to be contradicting or difficult to explain. The rescue data were not convincing.

---

## [Referee Report · Reviewer #3 (Public Review)]

Summary:

SMN expression in non-neuronal cells, particularly in limb mesenchymal progenitors is essential for the proper growth of chondrocytes and the formation of adult NMJ junctions.

Strengths:

The authors show copy numbers of smndelta7 in MPC influence NMJ structure.

Weaknesses:

Functional recovery by FAP transplantation is not complete. Mesenchymal progenitors are heterogeneous, and how heterogeneity influences this study is not clear. Part of the main findings to show the importance of SMN expression in non-neuronal cells is partly published by the same group (Kim et al., JCI Insight 2022). In the study, the authors used Dpp4(+) cells. The difference between the current study and the previous study is not so clear.

---

## [Author Response]

The following is the authors’ response to the original reviews.

**Reviewer #1**
Major comments:1. The authors conclude that the bone growth defects are chondrocyte-specific, highlighting no changes in the IGF pathway. However, other bone cells such as mesenchymal progenitors, osteoblasts, osteocytes, and marrow stromal cells are also lateral plate mesoderm derived and likely have roles in the bone growth phenotypes (a). Additionally, while the size decrease of the proliferative zone was stated, no actual proliferation assays such as BrdU were conducted (b). With the elements being of such small size in the mutants, the defects are likely to be found at the earliest stages of limb development at E11.5-E13.5 and may be due to mesenchymal to chondrocyte transitions or defects in osteoblast lineage development (c). Overall, the skeletal characterization is not rigorous and does not identify even a likely cellular mechanism. Further, a molecular mechanism by which SMN functions in mesenchymal progenitors, chondrocytes, or osteoblast lineage cells has not been assessed (d).

(a, c) As the reviewer commented, it seems to be a very important point to evaluate whether there is any problem in embryonic development from the time of mesenchymal cell condensation of the limb bud to the primary ossification center. However, when Hensel et al evaluated bone growth in P3 of severe SMA mice, the growth defect was not very large, with control femur length 3.5 mm and mutant 3.2 mm. it seems that even if SMN defects occur, there is no major problem with endochondral bone formation in the embryonic period (Hensel et al., 2020).

In this study, the SMN2 1-copy mutant with the bone growth defect was found to have a similar reduction in SMN protein to the severe SMA mouse model in experiments quantifying SMN protein. When Hensel et al. performed an in vitro ossification test on primary osteoblasts from the other severe SMA mouse model (Taiwanese severe SMA), they found no significant difference compared to controls. In femurs at P3 from severe SMA mice, they found no difference in bone voxel density and bone thickness (Hensel et al., 2020). In our data, bone thickness was not different in Figure 1 and Figure 1 – figure supplement 2, and BMD was actually greater. Thus, we believe that osteoblast and osteocyte function does not appear to be impaired by the absence of SMNs. When we looked at cortical osteoblasts in our new Figure 1-figure supplement 2, there did not appear to be a significant difference in density.

Furthermore, it is unlikely that BMSCs contributed to the bone growth we observed up to 2 weeks of age. the Lepr+Cxcl12+ BMSC population, which constitutes 94% ± 4% of CFU-F colonies formed by bone marrow cells (Zhou et al.k, 2014), is Prrx1-positive, and is known to be capable of osteogenesis in vivo, was only shown to differentiate into osteoblasts and form new bone in adults over 8 weeks of age. In the Lepr-cre; tdTomato; Col2.3-GFP mouse model, few cells expressing the osteoblast marker Col2.3-GFP are found before 2 months, and only about 3% of femur trabecular and cortical osteocytes express tdTomato at 2 months (Zhou et al., 2014). In Cxcl12-CreER; tdTomato; Col2.3-GFP mouse model, the researchers did not find tomato positivity in osteoblasts and osteocytes even after administration of tamoxifen at P3 and analysis 1 year later (Matsushita et al., 2020).

We, therefore, concluded that the bone growth abnormalities observed in SMN2 1-copy mutants are due to problems in endochondral ossification caused by chondrocyte defects and not due to other Prrx1-lineage skeletal cells.

(b) According to the reviewer's suggestion, we evaluated cell proliferation in the new Figure 1J-L by performing immunostaining for the Ki67 proliferation marker in growth plates.

(d) As the reviewer pointed out, we enhanced the mechanism study and found the reduction of chondrocyte-derived IGF signaling and hypertrophic marker in new Figure 2. We evaluated the density of osteoblasts and osteoclasts, which can affect bone mineralization. We highlighted the limited impact of BMSCs on bone growth in the first two weeks of life. In a previous study, SMN-deleted osteoblasts did not show any issues with ossification (Hensel et al., 2020). In fact, osteoblast density in the SMN2 1-copy mutant was not different from the control, indicating that the skeletal abnormalities can largely be attributed to deficiencies in endochondral ossification caused by chondrocytes. Since chondrocytes are the local source of IGF and our mutants exhibit phenotypes similar to mouse models with reduced IGF, such as downregulated expression of Igf1 and Igfbp3, downregulated IGF-induced hypertrophic gene expression, reduced AKT phosphorylation, proliferation, and growth plate zone length, SMN-deleted chondrocytes probably showed these phenotypes due to decreased IGF secretion. Now, we added new Figure 2A-C, and E.

1. Is the liver the only organ/tissue that supplied IGF to the chondrocytes or are other lateral plate mesoderm-derived cells potential suppliers? It's not possible to pin SMN deletion in chondrocytes as intrinsic ignoring the other bone cell types that it is depleted from in the Prrx1Cre genetic model.

Recently, Oichi et al. reported that the local IGF source in the growth plate is chondrocytes by in situ hybridization and p-AKT staining (Oichi et al., 2023). When we measured IGF in chondrocytes isolated from articular cartilage, the expressions of Igf1 andIgfbp3 were markedly reduced in chondrocytes with SMN deletion compared to controls (New Figure 2E), suggesting that intrinsic SMN expression in chondrocytes plays an important role in the growth plate.

1. Why is SMN protein being isolated from FAPs to assess levels in the null/SMN2 single copy/double copy mutants when the bone defects are supposed to be a chondrocyte-specific phenotype? This protein expression needs to be confirmed in chondrocytes themselves, and or other Prrx1Cre lineaged skeletal cells.

According to the reviewer’s suggestion, we attempted to evaluate the protein levels in chondrocytes of the SMN2 1-copy mutant. However, we were unable to obtain sufficient numbers of chondrocytes, because of poor proliferation of mutant chondrocytes compared to controls in culture conditions. We could obtain ~10^4 viable cells from 1 mouse of SMN2 1-copy mutant. Therefore, our only options for confirming SMN deletion in chondrocytes were DNA and RNA work. As in the Prrx1-lineage FAPs that the amount of SMN protein correlates with the expression levels of full-length SMN mRNA (Figure 2H-J), we expect that the SMN protein in chondrocytes would be fully depleted due to poor full-length SMN mRNA expression (Figure 2H).

1. Figure 2E should have example images of each type of NMJ characterization.

We revised our figure by adding the example images in new Figure 3E.

1. What are the overall NMJ numbers in the normal formation period? Are these constant into the juvenile period when the authors say the deterioration occurs?

We appreciate the reviewer's constructive comments, and it would be interesting to see if we could see a difference in the total number of NMJs. However, there is one NMJ in every myofiber, and each muscle has hundreds to thousands of myofibers. The technical difficulty of confocal imaging an entire muscle, which can be several millimeters across, precludes experiments that count every NMJ and show a difference. It may be possible to do so by combining clearing and confocal line scanning techniques. In our analysis of the NMJ, the formation of the NMJ in the mutant appears to be normal. Additionally, the number of myofibers seems to be the same, and there may be no difference in the total NMJ number.

1. For transplantation experiments the authors sorted YFP or TOMATO+ cells from the Prrx1Cre mice muscles, but refer to them as FAPs. It is known that other cells including tenocyte-like cells, pericytes, and vascular smooth muscle cells are identified by this reporter line. Staining for TOMATO colocalization with PDGFRA would help to clarify this.

In the method ‘Hindlimb fibro-adipogenic progenitors isolation’ section, we sorted 7AAD–Lin–Vcam–Sca1+ population refers to FAPs. For FAPs transplantation, we also used YFP or TOMATO+ FAPs (7AAD–Lin–Vcam–Sca1+). The ‘FAPs transplantation’ method section did not specify the FAPs population in detail. This has been fixed in the new method. Sca1 (Ly6a) is an effective marker for identifying FAPs within Prrx1-lineage cells, as well as Pdgfra (Leinroth et al., 2022).

1. The authors only compare the SMN2 single copy mutant transplantation to contralateral to show rescue, but how does this compare to overall wt morphology?

According to the reviewer’s constructive comment, we compared them with wild-type morphology (new Figure 7A-D).

1. The asterisks of TOMATO+ in Figure 6A are confusing. FAPs do not usually clump together to form such large plaques and are normally much thinner tendrils. What is the reason for this?

As the reviewer states, FAPs have a fibroblast-like morphology with elongated thinner tendrils. The Figure 6A image in the figure shows a Z-sliced cell body portion of FAP, where the nucleus is located, and it appears blunt. We attached imaged tomato+ FAPs, in which their cell body parts are plaque-like.

**Author response image 1. sa4fig1:** Tomato+ FAPs in muscle.

1. Would transplantation of healthy FAPs after NMJ maturation in SMN mutants still rescue the phenotype? Assessment of this is key for therapy intervention timelines moving forward.

It will be very interesting to see if the phenotype improves after NMJ maturation by healthy FAPs transplantation, but this is a technically difficult experiment to do because we found that FAPs do not implant effectively when injected into naive adult muscle. The transplantation into the adult is sufficiently possible if accompanied by an injury, but this eventually leads to new formation of NMJ again. Thus, it seems impossible to do transplantation experiment after NMJ maturation through general methods. If we discover a method to efficiently rescue SMNs from FAPs or identify a factor that affects FAPs' influence on NMJ, then we may be able to conduct this experiment.

Reference

Hensel, N., Brickwedde, H., Tsaknakis, K., Grages, A., Braunschweig, L., Lüders, K. A., Lorenz, H. M., Lippross, S., Walter, L. M., Tavassol, F., Lienenklaus, S., Neunaber, C., Claus, P., & Hell, A. K. (2020). Altered bone development with impaired cartilage formation precedes neuromuscular symptoms in spinal muscular atrophy. Human Molecular Genetics, 29(16), 2662–2673. https://doi.org/10.1093/hmg/ddaa145

Leinroth, A. P., Mirando, A. J., Rouse, D., Kobayahsi, Y., Tata, P. R., Rueckert, H. E., Liao, Y., Long, J. T., Chakkalakal, J. V., & Hilton, M. J. (2022). Identification of distinct non-myogenic skeletal-muscle-resident mesenchymal cell populations. Cell Reports, 39(6), 110785. https://doi.org/10.1016/j.celrep.2022.110785

Matsushita, Y., Nagata, M., Kozloff, K. M., Welch, J. D., Mizuhashi, K., Tokavanich, N., Hallett, S. A., Link, D. C., Nagasawa, T., Ono, W., & Ono, N. (2020). A Wnt-mediated transformation of the bone marrow stromal cell identity orchestrates skeletal regeneration. Nature Communications, 11(1). https://doi.org/10.1038/s41467-019-14029-w

Oichi, T., Kodama, J., Wilson, K., Tian, H., Imamura Kawasawa, Y., Usami, Y., Oshima, Y., Saito, T., Tanaka, S., Iwamoto, M., Otsuru, S., & Enomoto-Iwamoto, M. (2023). Nutrient-regulated dynamics of chondroprogenitors in the postnatal murine growth plate. Bone Research, 11(1). https://doi.org/10.1038/s41413-023-00258-9

Zhou, B. O., Yue, R., Murphy, M. M., Peyer, J. G., & Morrison, S. J. (2014). Leptin-receptor-expressing mesenchymal stromal cells represent the main source of bone formed by adult bone marrow. Cell Stem Cell, 15(2), 154–168. https://doi.org/10.1016/j.stem.2014.06.008

**Reviewer #2**
Major comments:1. Regarding bone deficits - CT analysis of bones should be more comprehensive than Figure 1A shows. How about cross-sections? (a) Are bone phenotypes also age-dependent? (b) PCR was done only for SMA and related proteins (such as IGF). IGF protein in the blood and relevant organs should be studied. Why not include biomarkers of osteoblasts or/and osteoclasts and their regulators? (c)

(a) We appreciate the reviewer’s constructive comment. we added longitudinal section views in new Figure 1A and a description of trabecular bone volume and secondary ossification center in the main text.

(b) Age-dependent evaluation is an important point. By adulthood, the difference between the SMN2 1-copy mutant and the control is much larger, and even at birth there is a slight difference, although not as large as at 2 weeks of age. We focused our phenotyping on bone growth at 2 weeks of age, a time when new bone formation by BMSCs is less influential, when bone growth is primarily driven by endochondral ossification of chondrocytes, and before the defect in theNMJ is primarily manifested.

(c) As the reviewer comments, it is important that IGF are evaluated in tissues other than liver. However, the liver is most likely the source of systemic IGF, as shown by the liver-specific deletion of Igf1 and knockout of Igfals, a protein that forms the IGF ternary complex, which is predominantly expressed in the liver. This resulted in a 90% drop in serum IGF levels and a phenotype of shortened femur length and growth plates in the double KO mice (Yakar et al., 2002).

The local IGF source in the growth plate is chondrocytes confirmed by Igf1 in situ hybridization and p-AKT staining (Oichi et al., 2023). From the In situ hybridization data, we can observe that bone marrow and bone do not express Igf1 at all, but only perichondrium and chondrocytes in the resting zone express Igf1 mRNA. Therefore, we can see that the only supplier of IGF among LPM-derived cells is chondrocytes, and in the new figure 2, we measured IGF pathway expression and AKT phosphorylation in chondrocytes. We have confirmed that the expression of Igf1/Igfbp3 is reduced in chondrocytes with SMN deletion.

To assess serum IGF level, we could not set up this experiment condition during our revision period due to the requirement of administrative procedures for purchasing new apparatuses and the limitation of our research funds. However, as previously stated, there is no difference in the expression of Igf1 and Igfals in the liver, which accounts for 90% of serum IGF levels. Therefore, we did not anticipate significant variations in serum IGF levels.

Evaluation of osteoblasts or osteoclasts was done by section staining due to sampling difficulties for PCR. we assessed osteoblasts and osteoclasts state in new Figure 1-figure supplement 2.

1. What is the relationship between deficits of bone deficits and muscle deficits or even NMJ deficits? Are they inter-related? Is skeletal muscle development also defective in Smn∆MPC mice? Can NMJ deficits result from bone deficits? Or vice versa?

Unfortunately, the reviewer's comments are very difficult to clarify in our study using the Prrx1-cre model. In skeletal muscle development, the myofiber number was not significantly different in our mouse models. A study has shown that inactivating noggin, a BMP antagonist expressed in condensed cartilage and immature chondrocytes, results in severe skeletal defects without affecting the early stages of muscle differentiation (Tylzanowski et al., 2006). Therefore, bone may not have a significant impact on the early development of muscle, but later in postnatal development it may have an impact on motor performance issues. The relationship between bone and NMJ hasn't been studied. The impact of bone defects on motor skill may result in muscle weakness and NMJ problems. In our study, we showed that NMJ deficit rescue by transplantation of FAPs and decreased IGF in chondrocytes, a key source of local IGF. This suggests that the functions of FAPs in NMJ and chondrocytes in bone deficit are crucial, rather than each other's influence.

1. Regarding the rescue experiment, the interpretation of the data should be careful. Evidently, healthy FAPs (td-Tomato positive) were transplanted into TA muscles of 10 days-old SMN2 1-copy SmnΔMPC mice, and NMJs were looked at P56. The control was contralateral TA that was injected with the vehicle. As described above, the data had huge SEM and were difficult to interpret or believe. The control perhaps was wrong if FAPs act by releasing "chemicals" because FAPs from one leg may go to other muscles via blood. Second, if FAPs act via contact, the data shown did not support this. Two red FAPs were shown in Figure 6, one of which was superimposed with a nerve track to one of the three NMJs. This NMJ however did not show any difference to the other two, which did not support a contact mechanism. These rescue data were not convincing.

We appreciate the reviewer’s critical comment, but the reviewer appears to have confused the minimum and maximum range bars in the box-and-whisker plot with the SEM error bar in the bar graph. We apologize for the insufficient description of the figure legends section. We revised them. New Figure 7C, which is a bar graph, has a sufficiently short SEM error bar. In contrast, box-and-whisker plots B and D depict the minimum and maximum range, instead of the SEM, and they are significantly different with a p-value of less than 0.001. If FAPs affect the NMJ via a paracrine factor or ECM with a short range of action, they may rescue the NMJ defect in a non-contact-dependent manner, without affecting the contralateral muscle. Also, the FAPs are heterogeneous, so if only a certain subpopulation rescues, the tomato+ FAP in the figure may not be the rescuing cells.

1. For most experiments, the "n" numbers were too small. 3-5 mice were used for bone characterization. For the NMJ, most experiments were done with 3 mice. It was unclear how many NMJs were looked at. Perhaps due to small n numbers, the SEM values were enormous (for example, in Figure 6).

As with the response to the previous comment, this is due to confusion between box-and-whisker plots and bar graphs, and our data was determined to be significant using the appropriate statistical method.

1. Also for experimental design, some experiments included four genotypes of mice (Fig. 1 J,K) whereas some had only three (Fig.1 A, B, C, D and Fig.3) and others had two (many other figures).

In the first experiments to confirm the phenotypes, we tested the 2-copy mutant, but it was not significantly different from the wild type, and in subsequent experiments, we mainly tested the only 1-copy mutant.

1. What was the reason why mixed muscles were used for NMJ characterization (TA versus EDL)? Why not pick a type I-fiber muscle and a type II-fiber muscle?

We appreciate the constructive comment from the reviewer. Firstly, we conducted a phenotype analysis on the TA muscle. For electrophysiological recording, the EDL muscle should be used for intact nerve with muscle preparation, technically. Additionally, for TEM imaging, EDL was a suitable muscle to locate NMJ positions before TEM processing. Both TA and EDL muscles are adjacent and have similar fiber-type compositions. It would be important to observe in different fiber types of muscles, but when we first identified the phenotype, various types of limb muscles showed similar defects, so we focused on specific muscles.

1. The description of mouse strains was confusing. SMN2 transgenic mice (with different copies) were not described in the methods.

We apologize for the insufficient description of the method section. By crossing mice with the SMN2+/+ homologous allele, SMN2 heterologous mice with only one SMN2 allele are SMN2 1-copy mice (SMN2+/0) and SMN2 homologous mice are SMN2 2-copy mice (SMN2+/+). We revised our manuscript method ‘Animals’ section.

ReferenceOichi, T., Kodama, J., Wilson, K., Tian, H., Imamura Kawasawa, Y., Usami, Y., Oshima, Y., Saito, T., Tanaka, S., Iwamoto, M., Otsuru, S., & Enomoto-Iwamoto, M. (2023). Nutrient-regulated dynamics of chondroprogenitors in the postnatal murine growth plate. Bone Research, 11(1). https://doi.org/10.1038/s41413-023-00258-9

Tylzanowski, P., Mebis, L., and Luyten, F. P. (2006). The noggin null mouse phenotype is strain dependent and haploinsufficiency leads to skeletal defects. Dev. Dyn. 235, 1599–1607. doi: 10.1002/dvdy.20782

Yakar, S., Rosen, C. J., Beamer, W. G., Ackert-Bicknell, C. L., Wu, Y., Liu, J. L., Ooi, G. T., Setser, J., Frystyk, J., Boisclair, Y. R., & LeRoith, D. (2002). Circulating levels of IGF-1 directly regulate bone growth and density. Journal of Clinical Investigation, 110(6), 771–781. https://doi.org/10.1172/JCI0215463

**Reviewer #3**
1. The authors used Prrx1Cre mouse with floxed Smn exon7(Smnf7) mouse carrying multiple (one or two) copies of the human SMN2 gene. Is it expressed both in chondrocytes and mesenchymal progenitors in the limb?

We appreciate the reviewer's comment. We analyzed the deletion of Smn in chondrocytes and FAPs via Cre using genomic PCR and qRT-PCR, as depicted in new Figure 2. The SMN2 allele, which is expressed throughout the body, can rescue Smn knockout mouse lethality (Monani et al., 2000). Indeed, the short limb length and lethality observed in SMN2 0-copy mutants were mitigated by the presence of multiple copies of SMN2. Therefore, both Chondrocytes and FAPs may express SMN2 transcripts from the transgenic SMN2 allele.

1. Page 10 regarding Fig.2E, please show pretzel-like structure. In Figure 2E, plaque, perforated, open, and branched are shown; however, the pretzel is not shown. The same issue is for the Fig. 3D explanation in the text on page 12.

We appreciate the reviewer's constructive feedback. We included illustrative figures of all types of NMJ characterization, and the branched type is identical to the pretzel type. Therefore, we have replaced ‘branched’ with ‘pretzel’ in our text and revised Figure 3E by incorporating the example images.

1. The explanation of the electrophysiology for Fig.4 in the text on pages 12 and 15 (RRP) is not so convincing for the readers. It is advisable to add TEM data for transplantation if it is not technically difficult.

We appreciate the reviewer's critical feedback. Because we did not measure RRP directly, we removed speculation about the possibility of RRP difference. If observing the active zone with TEM and the docking synaptic vesicle would help quantify RRP, it is technically difficult to obtain images of sufficient quality to distinguish the active zones with our current TEM imaging technique.

1. The authors used the word FAP for 7AAD(-)Lin(-)Vcam(-)Sca1(+). It is recommended to show the expression of PDGFR alpha. Furthermore, as the authors stated in the text, mesenchymal progenitors (FAPs) are heterogeneous. Please discuss this point further. Other reports show at least 6 subpopulations using single-cell analyses (Cell Rep. 2022).

In the report, Ly6a (Sca1) is a good marker for FAPs, as well as Pdgfra (Leinroth et al., 2022). The 6 subpopulations expressed Ly6a. The one of subpopulations associated with NMJ was discovered. This population expressed Hsd11b1, Gfra1, and Ret and is located adjacent to the NMJ and responds to denervation, indicating an increased possibility of interaction with the NMJ organization. In further our study, we aim to determine which subpopulations are crucial for NMJ maturation by transplanting them to mutants for rescue.

1. How do authors determine the number of FAP cells for transplantation?

The FAPs transplantation was performed according to a previously reported our study (Kim et al., 2021).

ReferenceKim, J. H., Kang, J. S., Yoo, K., Jeong, J., Park, I., Park, J. H., Rhee, J., Jeon, S., Jo, Y. W., Hann, S. H., Seo, M., Moon, S., Um, S. J., Seong, R. H., & Kong, Y. Y. (2022). Bap1/SMN axis in Dpp4+ skeletal muscle mesenchymal cells regulates the neuromuscular system. JCI Insight, 7(10). https://doi.org/10.1172/jci.insight.158380

Leinroth, A. P., Mirando, A. J., Rouse, D., Kobayahsi, Y., Tata, P. R., Rueckert, H. E., Liao, Y., Long, J. T., Chakkalakal, J. V., & Hilton, M. J. (2022). Identification of distinct non-myogenic skeletal-muscle-resident mesenchymal cell populations. Cell Reports, 39(6), 110785. https://doi.org/10.1016/j.celrep.2022.110785

Monani, U. R., Sendtner, M., Coovert, D. D., Parsons, D. W., Andreassi, C., Le, T. T., Jablonka, S., Schrank, B., Rossol, W., Prior, T. W., Morris, G. E., & Burghes, A. H. M. (2000). The human centromeric survival motor neuron gene (SMN2) rescues embryonic lethality in Smn(-/-) mice and results in a mouse with spinal muscular atrophy. Human Molecular Genetics, 9(3), 333–339. https://doi.org/10.1093/hmg/9.3.333